# Spatiotemporal coordination of reovirus peripheral core replication to perinuclear whole virus assembly

Justine Kniert[1], Dante Terino[1], Heather E. Eaton[1], Qi Feng Lin[1], Shiau-Yin Wu[2], Hilmar Strickfaden[2], Maya Shmulevitz[1]*

**1** Department of Medical Microbiology and Immunology, Li Ka Shing Institute of Virology, University of Alberta, Edmonton, Alberta, Canada, **2** Cell Imaging Core, Faculty of Medicine and Dentistry, University of Alberta, Edmonton, Alberta, Canada

\* shmulevi@ualberta.ca

## Abstract

Reoviruses coordinate their replication and assembly through intricate spatial and temporal compartmentalization within host cells. In this study, we elucidate the dynamics of mammalian orthoreovirus (reovirus) core replication and viral particle assembly. Using high-resolution immunofluorescence confocal microscopy, we tracked input cores and de novo cores, revealing that input cores initially form peripheral, OC-negative factories that migrate inward while seeding independent peripheral factories. Over time, these input factories transition into intermediate core-plus-outercapsid (OC) factories, which are essential for full virion assembly in the perinuclear region. Notably, de novo core proteins predominantly form independent peripheral factories that can merge or mix with others, resulting in interconnected networks. We further demonstrate that microtubules are dispensable for early core movement and factory formation but are crucial for the transition of mature, assembled virions into perinuclear deposits and for timely virion production. Disruption of microtubules delays full virus assembly, reducing progeny yield. Our findings reveal a complex, regulated interplay between spatial organization and cytoskeletal components during reovirus infection, providing insights into mechanisms that could be targeted for antiviral interventions.

## Author summary

Reoviruses are a widely studied group of viruses that serve as a valuable model for understanding viral replication and assembly. Due to their well-characterized biology and minimal pathogenicity, reoviruses are also being explored as therapeutic agents in cancer and vaccine development. Despite this, the detailed mechanisms by which reoviruses coordinate their replication and assemble new infectious particles inside host cells remain incompletely understood. In our

**Data availability statement:** All relevant data are within the manuscript and its Supporting Information files.

**Funding:** This publication is supported through project grants to MS from the Canadian Institutes of Health Research (CIHR, grants #159594, #180277, and #196976), the Natural Sciences and Engineering Research Council of Canada (NSERC, grant #RGPIN2022-03418), a salary award to MS from the Canada Research Chairs (CRC), infrastructure support to MS from the Canada Foundation for Innovation (CFI) and a generous donation by Linda M. Youell in memory of her husband, Gerry. JK received scholarships from the University of Alberta Faculty of Medicine and Dentistry (FoMD), the Li Ka Shing Institute of Virology (LKS) and the Alberta Province. DT received a scholarship from the University of Alberta. QFL received scholarships from the CIHR, the NSERC, the FoMD, the LKS, and the Cancer Research Institute of Northern Alberta (CRINA). The funders had no role in study design, data collection and analysis, decision to publish, or preparation of the manuscript.

**Competing interests:** The authors have declared that no competing interests exist.

study, we used advanced imaging techniques to track individual viral components and factories within infected cells over time. We discovered that initial input cores form distinct peripheral factories that move inward, where they acquire outercapsid proteins and assemble into full viral particles in a regulated, spatially organized process. Microtubules, a key part of the cell's skeleton, are not needed for early assembly steps but are essential for transporting mature viruses into the cellular center. Our findings shed light on the complex yet highly coordinated journey of reoviruses as they replicate, assemble, and spread, offering potential avenues for antiviral strategies and improving the design of reovirus-based therapies.

## Introduction

Mammalian orthoreovirus 3 (reovirus) is one of the most well-studied members of the *Reoviridae* family. Collectively, *Reoviridae* infect almost all eukaryotic kingdoms, often presenting pathological challenges to host species such as grass carp reovirus in fish, rice dwarf virus in rice, bluetongue virus in ruminants, and rotavirus in humans [1–5]. In contrast, reovirus infects the majority of humans before adolescence, yet causes minimal-to-no disease [6–12]. Although reovirus naturally causes enteric and respiratory infections [13–16], it is also capable of replicating in transformed cells [17–22] and is currently undergoing clinical trials as a candidate oncolytic therapy for a myriad of cancers.

Reovirus is a segmented, double-stranded RNA (dsRNA) virus composed of two capsid layers. The outermost layer is composed of the outercapsid (OC) proteins σ1, σ3, and μ1. The OC coats the inner core particle, which is comprised of five proteins: σ2 and λ1 form the core shell; λ2 pentamers form turrets at each of the 12 vertices of the icosahedral viral particle; and λ3 RNA-dependent RNA polymerases and μ2 co-factors reside inside the core at each vertex. Cell attachment and entry of reovirus is mediated by the σ1 protein, which engages cell surface sialic acids and junctional adhesion molecule A [23], followed by binding of the core turret protein λ2 to cellular β-integrins to initiate endocytosis [24,25]. Within the lysosome, OC proteolysis promotes penetration of lysosomal membranes to deposit intact core particles into the cytoplasm [26–33]. Cores then initiate replication, which proceeds in membraneless pseudo-organelles formed through phase separation (also referred to as condensates, replication organelles, or herein, factories). Factory structures are formed by non-structural core-binding μNS and RNA-binding σNS proteins, tubulin-binding core protein μ2, and a variety of host factors including endoplasmic reticulum fragments and associated ribosomes, cytoskeletal elements, and host chaperone proteins [34–44]. Within the core particle, the viral polymerase λ3 and NTPase μ2 synthesize (+)-sense RNAs from dsRNA genomes, and transcripts are subsequently capped by λ2 as they exit the λ2 turrets [45–48]. Viral proteins are then translated via host machinery. Core proteins, together with (+)-sense RNAs, assemble into *de novo* cores that continue to amplify the replication process. Whole virions are generated

as OC proteins coat existing core particles prior to eventual egress. In summary, the core particle is the transcriptionally active unit for replication, while the whole virus is the infectious particle due to the presence of OC proteins.

To maintain productive amplification, replication-competent core particles cannot become immediately coated by OC proteins. Recent studies demonstrated that reovirus delays OC assembly by spatiotemporal segregation of core amplification and OC assembly [49]. Although all core and OC proteins are synthesized simultaneously, during early infection (~8 hpi), transcriptionally active core-only factories form in the periphery, while OC proteins are segregated to lipid droplets (LDs). The exclusion of OC proteins from core-only peripheral factories permits logarithmic amplification of *de novo* cores without premature full assembly and transcriptional cessation. As infection proceeds (~12 hpi), factories accumulate in intermediate regions between the periphery and nucleus, adjacent to OC-associated LDs, where OC proteins start to accumulate within the factories. Lastly (>16 hpi), large perinuclear factories form where whole virions accumulate in paracrystalline arrays. Altogether, reoviruses employ four distinct compartments for replication and assembly, including peripheral core-only factories, OC-associated LD compartments, intermediate core-plus-OC factories and perinuclear deposit compartments for fully assembled viruses. This spatiotemporal compartmentalization allows reovirus cores to amplify replication in core-only peripheral factories, while whole virions are segregated away in intermediate and perinuclear factories [49].

The current study investigated mechanisms governing the formation and dynamic transitions of reovirus replication and assembly compartments. Using innovative approaches to label and modify core proteins in combination with immunofluorescence confocal microscopy (IF-CM), it was determined that input cores initially establish core-only factories at the periphery. As infection progresses, input core factories transition into intermediate and perinuclear areas. Simultaneously, *de novo* core-only factories lacking input cores continually emerge at the periphery. Through distinguishing input cores from *de novo*-produced proteins, it was found that independent input cores primarily seed independent compartments, whereas *de novo* products mix freely in later generations of core-only peripheral factories. To determine what forces may drive OC(-) versus OC(+) factory movements, investigations turned to microtubule (MT) interactions. While prior findings suggest that MTs are involved in cell entry [50], factory formation and organization [51–54], genome packaging [55], and movement [56], these studies focused on late stages of replication and were conducted before the discovery of distinct replication and assembly compartments. Under nocodazole treatment to disrupt MTs, core-only factories and their transition to intermediate compartments were unaffected, suggesting MT independence. Further factory progression was abolished, preventing deposition of fully assembled virions in perinuclear regions. Additionally, kinetic analysis revealed approximately a 10-fold delay in whole virus production following MT disruption suggesting that MTs contribute to the efficiency and timeliness of OC assembly into fully assembled viruses. Altogether, the findings reveal complex dynamic formations and transitions between four compartments to orchestrate reovirus core amplification, outercapsid assembly, and deposition of fully assembled virions. These findings underscore the intricate strategies viruses use to optimize genome amplification and assembly.

## Results

### Input cores initially form peripheral core-only factories then transition into the perinuclear core-plus-OC factories while de-novo core-only peripheral factories are formed devoid of input cores

Recent findings reveal that reovirus replication does not occur within single, homogeneous factories. Instead, peripheral core-only factories initially form, containing core and non-structural proteins, but lacking outercapsid (OC) proteins, which are instead co-localize with LDs. These core-only factories permit amplification of cores away from OC proteins, which could otherwise cause premature full assembly and, consequently, halt further amplification. As replication proceeds, intermediate mid-cytoplasmic factories also form, where cores and LD–docked OC proteins mix. Finally, fully assembled and inert virions are deposited at large perinuclear compartments. Previous studies also found that the factory-forming

non-structural μNS protein forms inclusion bodies in cells when transfected alone [34,51]. When cells with pre-existing μNS inclusion bodies were infected in the presence of cycloheximide, which thwarts viral replication, incoming cores co-localized with the μNS inclusion bodies [36]. These findings raised the possibility that incoming cores might reside in the early peripheral core-only factories. Alternatively, factories could be established at locations distant from input cores, using cytoplasmic viral proteins and RNAs produced by input cores. To distinguish between these possibilities, we aimed to develop a strategy to monitor input core localization throughout a bona fide reovirus replication cycle. To this end, core particles were generated in vitro from reovirus (strain T3D^PL) by chymotrypsin-mediated removal of OC proteins. The cores were then labelled with Alexa Fluor dyes. SDS-PAGE analysis confirmed that in vitro cores devoid of OC proteins were generated and efficiently fluorophore labelled (S1A Fig). Although core particles lack the OC machinery needed for cellular binding and entry events, they can still be effectively encapsidated in lipid-based transfection reagents and delivered into cells [52] (S1B Fig). To establish if Alexa Fluor labelled cores (AF-cores) had comparable replication kinetics to non-labelled cores, AF- and non-labelled cores were transfected into H1299 cells and de novo virus titers were quantified by plaque titration. While new virus production during whole virus infection is normally seen at 4–6hpi, transfected cores likely have accelerated onset of infection. Beyond the early timepoints when titers are close to the limit of detection, AF-cores demonstrated growth kinetics similar to those of non-labeled cores (S1C Fig), supporting their utility in tracking input core fate during infection.

To evaluate whether incoming cores reside in the early peripheral core-only factories, IF-CM was used to monitor whether AF-cores transfected into cells colocalized with the μNS-positive but OC-negative core-only factories. H1299 cells were transfected with AF647-cores, fixed at 6, 9, and 12 hours (S2A, C Fig) post-transfection (hpt), immunostained with antibodies directed against μNS and OC σ3 proteins, and subjected to IF-CM analysis (Fig 1A, B). It should be noted that σ3 immunostaining was used to identify OC-containing factories, and herein "OC" often refers to σ3. Three-dimensional images were then processed using the Volocity software for quantification as described in the Methods section. Aggregates of AF647-cores were sometimes visible (Fig 1, demarcated with asterisks), but such aggregates became in-focus at z-planes corresponding to the surface of cells, and were never associate within intracellular virus protein-positive factories; accordingly, such aggregates were excluded from analysis." In the context of natural infection, 'early' infection refers to ~4–9 hpi, 'mid' infection to ~9–12 hpi, and 'late' infection to ~12–16 hpi, with titer saturation occurring around 15 hpi [49]. However, in experiments utilizing transfection of AF-core particles, the binding and uncoating steps of normal infection are bypassed, and thus, infection by AF-cores is accelerated, with saturation reached by 9–12 hpt. At an early time point of 6hpt, ~70% of all μNS-positive OC-negative factories were also positive for AF-cores; this suggested that the majority of initial peripheral core-only factories form around input cores (Fig 1C).

Having determined that input cores reside in the early peripheral core-only factories during bona fide reovirus replication, it became possible to determine whether early core-only factories move towards the nucleus to become the larger mid-cytoplasmic factories, or whether input cores remain at the periphery while larger mid-cytoplasmic factories form independently. To distinguish between these possibilities, analysis of AF-core location over time was conducted. During early infection (6hpt), AF-cores were predominantly in OC(-) factories (Fig 1B top, and C), located farthest from the nucleus (Fig 1D) and were small in volume (Fig 1E). As time progressed, more AF-cores accumulated in OC-containing factories (Fig 1B bottom, and 1C) found closer to the nucleus (Fig 1D), which were larger than their OC(-) counterparts (Fig 1E) and constituted the majority of the viral factory volume by 12hpt (Fig 1F). These findings suggest that input cores initially start at peripheral core-only factories, but transition over time into mid-cytoplasmic OC-containing factories. Moreover, the population of OC(-) factories lacking input cores increased significantly in number (Fig 1G) and volume (Fig 1H) between 6–12hpt, suggesting that new de novo factories devoid of input cores are formed at the periphery of cells as infection process. Altogether, the data suggest that initial peripheral OC(-) factories are formed predominantly around input cores, which transition into the nuclear-proximal factories over time, while new de-novo OC(-) factories devoid of input cores are seeded in the periphery (Fig 1I).

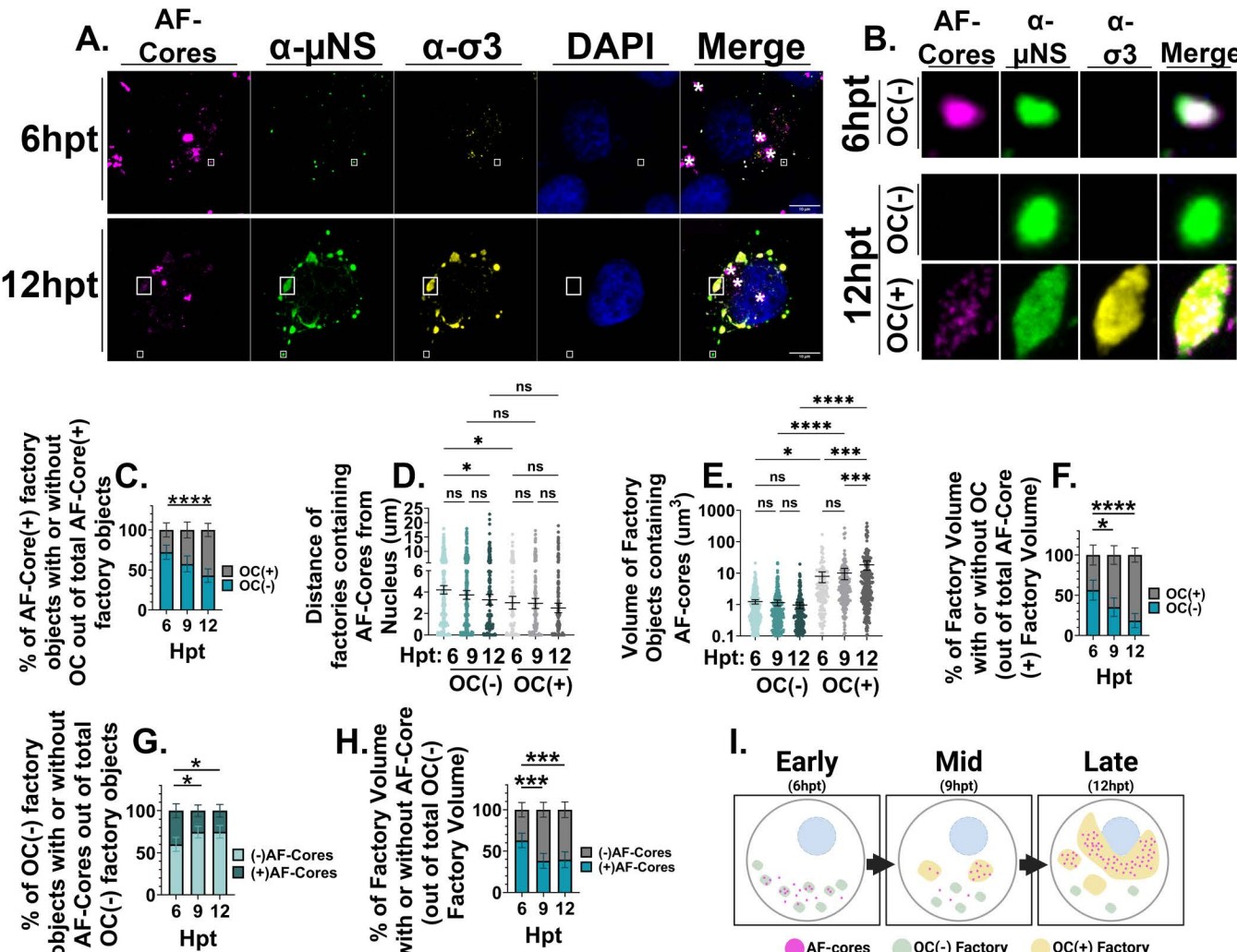

**Fig 1. Input cores initially form peripheral core-only factories then transition into the perinuclear core-plus-OC factories while *de novo* core-only peripheral factories are formed devoid of input cores.** H1299 cells were transfected with ~1000 AF 647-labelled reovirus core particles per cell (magenta). At 6, 9 and 12 hpt cells were fixed and subsequently immunostained with rabbit polyclonal sera generated against µNS (µNS, AF 488, green), mouse monoclonal anti-σ3 (10G10, Cy3, yellow) and nuclei were stained with DAPI (blue). (A) Representative images captured at 6 and 12 hpt. Aggregates formed by AF-cores are denoted with a white asterisk in image panels. (B) Select regions of OC(-) and OC(+) factories were artificially blown-up using Adobe Photoshop (white boxes). All images were acquired via immunofluorescence spinning disk confocal microscopy. (C) Of the total AF-core containing factories within a cell, the percentage of factories with or without outercapsid staining were graphed Statistical analysis is reported as ordinary two-way ANOVA with Tukey's multiple comparisons test. (D) The edge-to-edge distance of each factory object with AF-cores, with or without OC staining, were quantified and graphed Statistical analysis is reported as ordinary one-way ANOVA with Sidak's multiple comparisons test. (E) The volume of AF-core containing factories. Statistical analysis is reported as ordinary one-way ANOVA with Sidak's multiple comparisons test. (F) Of the total factory volume containing AF-cores within a cell, the percentage of factory volume containing or not containing OC was graphed. Statistical analysis is reported as ordinary two-way ANOVA with Tukey's multiple comparisons test. (G) Of the total number of OC(-) factories within a cell, the proportion of factories containing AF-cores or not were graphed. Statistical analysis is reported as ordinary two-way ANOVA with Tukey's multiple comparisons test. (H) Of the total OC(-) factory volume within a cell, the percentage of factory volume containing or not containing AF-cores was graphed. Statistical analysis is reported as ordinary two-way ANOVA with Tukey's multiple comparisons test. (I) Cartoon schematic describing conclusions of this work. Data represents five-ten images per condition and is representative of four independent experiments. Created in BioRender. *Shmulevitz, M.* (2025) https://BioRender. com/o6xobjd. All data is plotted as the mean +/- 95% confidence intervals. ****$p < 0.0001$, ***$p < 0.001$, **$p < 0.05$, ns $> 0.05$.

## Input cores establish independent peripheral compartments

When transfecting cores with one type of AF-fluorophore, it could not be distinguished whether input cores form distinct peripheral factories or whether different input cores share the same factories. Since incoming virions might bring distinct mutations and given that the peripheral factories are the first sites of core amplification, early coalescence of incoming cores could have implications for the timing of cooperativity between incoming virions. To address this question, *in vitro*-generated core particles were differentially labelled with AF-546 or AF-647 dyes. H1299 cells were first transfected with AF-546 labelled cores, and then one hour later, the cells were transfected with AF-647 labelled cores before fixation at seven hours after the initial transfection (Figs 2 and S3). AF-cores were transfected separately to prevent their co-encapsidation in lipoplexes and thus artificially delivered into the same location. Cells were immunostained and analyzed by IF-CM (Fig 2A, B). Only ~10% of AF-core puncta were shared between the two input cores (Fig 2B "Both", C), with 90% being either AF-546–positive or AF-647–positive (Fig 2B "AF-546 Only" and "AF-647 Only", C). It should be noted that there was a higher proportion of AF-647-independent puncta than AF-546, possibly due to increased transfection efficiency. These results suggest that input particles establish predominantly independent compartments at the beginning of infection (Fig 2D).

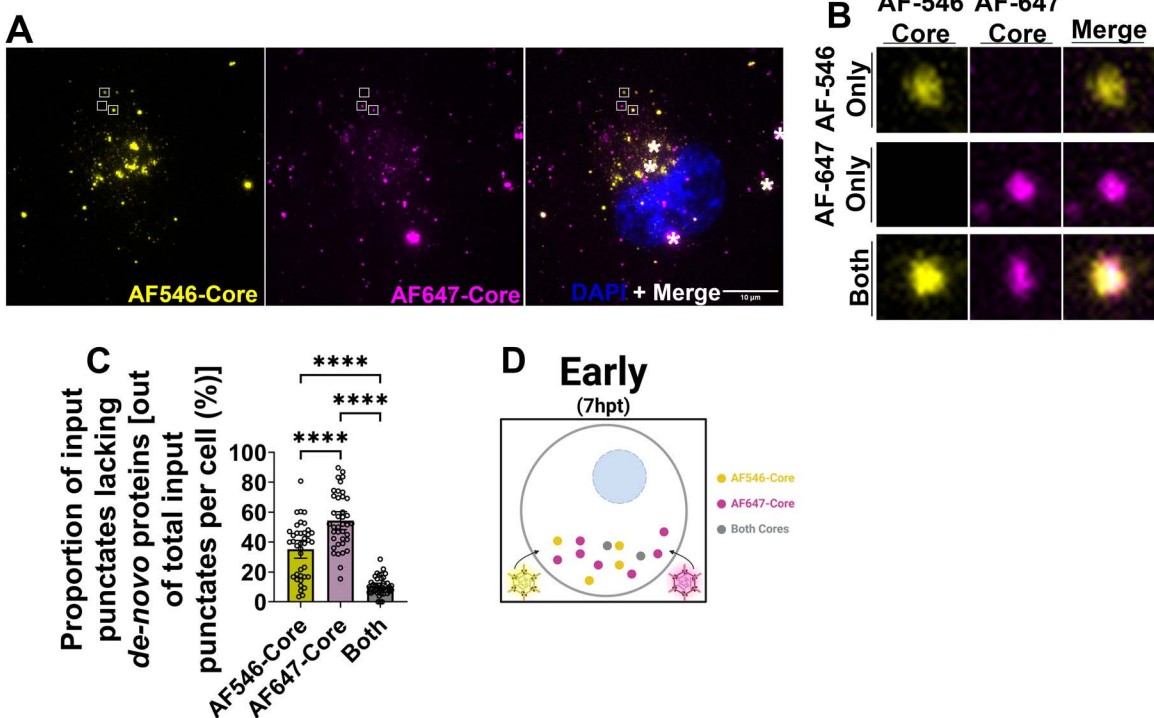

**Fig 2. Input cores establish independent peripheral compartments.** H1299 cells were first transfected with ~1000 AF-546 cores per cell. 1 hour later, the cells were transfected with ~1000 particles per cell of AF-647 labelled cores. 7 hpt, cells were fixed and processed for immunofluorescence confocal microscopy imaging. (A) Representative images of compressed Z-stacks. Cells were stained with DAPI to visualize nuclei. Aggregates formed by AF-cores are denoted with a white asterisk in image panels. (B) Compartments of every identity (AF-546 cores only, AF-647 cores only, and both cores) were selected and artificially blown-up using Adobe Photoshop (white boxes). (C) Using Volocity software (Quorum Technologies Inc.), the proportion of punctates containing AF546-cores alone, AF647-cores alone, or both cores co-localizing together was plotted. (D) Cartoon schematic summarizing findings, Created in BioRender. *Shmulevitz, M.* (2025) https://BioRender.com/j1js5vhData is representative of four independent experiments with a minimum of five cells per condition analyzed. Data is plotted as mean +/- SD. Statistical analysis is reported as ordinary one-way ANOVA between the mean of each column. ****p<0.0001, ***p<0.001, **p<0.05, ns>0.05.

### In early peripheral factories, most of the independent input core-containing OC(-) factories do not acquire newly synthesized core proteins from external factories

The finding that input cores initially produce independent peripheral compartments raised an interesting question: do *de novo* core proteins produced by an input core remain confined to their compartment of origin, or do *de novo* core proteins from independent input core factories intermix? Answering this question would provide insights into the potential for interaction and material exchange between distinct reoviral protein production centers. For example, if core proteins from different factories intermix, it could indicate a more interconnected network of viral replication sites, possibly allowing for resource sharing or adaptability in response to cellular defenses. However, addressing this question required us to create a new method for distinguishing core proteins expressed by distinct input cores, since there is a paucity of antibodies capable of discriminating between core proteins of distinct reoviruses, even among different strains or serotypes. Note that core proteins rather than OC proteins would need to be tracked because OC proteins are excluded from the core-only factories. To our knowledge, there are no examples of recombinant reoviruses containing exogenous sequences embedded into core proteins. To fill this gap, eight distinct flexible loops in the λ2 protein crystal structures were identified that may be amenable to insertion of FLAG and V5 epitope tags (Table 1 and S4 Fig, coloured sequences). The reovirus reverse genetics system was used to rescue recombinant viruses containing 2XFLAG or 1XV5 tags in each of the eight loops of λ2. Viruses were successfully recovered when 2XFLAG or 1XV5 tags were inserted at amino acids 1082 –1085, but not in the other seven loop regions (Table 1 and S4 Fig, orange sequences). We suspect this region is amenable to insertions because it does not hinder OC assembly and/or λ2 turret movement during transcription [46,48,53,54] (S4A–H Fig). The λ2 proteins of purified FLAG- and/or V5-tagged reoviruses (S4I Fig) could clearly be distinguished by Western blot analysis with corresponding tag-specific antibodies (Fig 3A). Similarly, cells infected by λ2-FLAG- and/or λ2-V5-tagged reoviruses could effectively be distinguished by immunofluorescence with FLAG- or V5- antibodies (Fig 3B). To determine whether the λ2-V5 and λ2-FLAG-tagged viruses had similar replication kinetics as the wild-type T3D$^{PL}$ reovirus, we evaluated the attachment efficiency to cells by flow cytometric analysis (Fig 3C), the expression of viral proteins over time by Western blot analysis (Fig 3D, E), and progeny virus production by plaque titration (Fig 3F). Though slightly attenuated in all three respects, the tagged viruses displayed replication kinetics similar to wild-type and were deemed appropriate models for application to studies of de novo protein distribution dynamics in factories.

While FLAG and V5 antibodies effectively stained de novo λ2-FLAG and λ2-V5 produced from their respective viruses, they did not stain incoming cores. This was likely because there were insufficiently exposed epitopes on incoming cores to be detected by the antibodies, or because the epitope is masked within the cores (S4C–E for whole core, S4F–H for whole virus structures). Given that incoming AF-647- and AF-546-labelled cores appeared to predominantly form independent peripheral factories (Fig 2), we asked whether, nevertheless, these peripheral factories shared *de novo* produced core proteins. In other words, could a λ2-FLAG product associate with an AF-λ2-V5-containing early peripheral

**Table 1. λ2 regions tested for insertion of tag sequences.**

| Amino Acid Region of λ2 (#) | Colour in S4 Fig | Virus recovered? |
|---|---|---|
| 441-449 | Yellow | No |
| 453-462 | Green | No |
| 1029-1032 | Dark Blue | No |
| 1046-1051 | Light Blue | No |
| **1082-1085** | **Orange** | **Yes** |
| 1154-1159 | Magenta | No |
| 1170-1175 | Purple | No |
| 1180-1190 | Red | No |

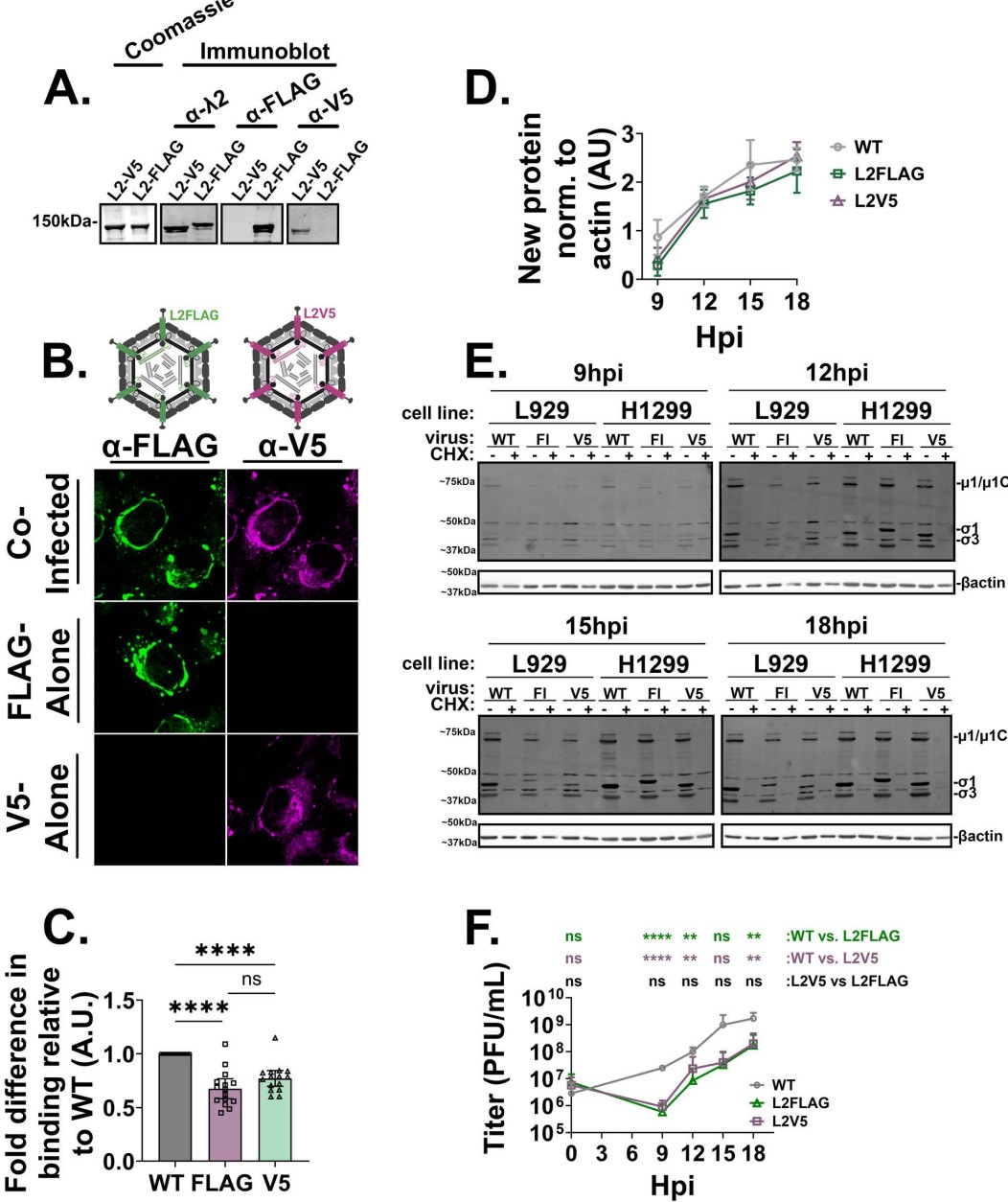

**Fig 3. Recombinant tagged reoviruses λ2-FLAG and λ2-V5 are comparable in their replication kinetics**. (A) Equivalent amounts of CsCl purified λ2-tagged viruses were subject to SDS-PAGE followed by either Coomassie blue staining or Western blot with polyclonal rabbit anti-λ2 (AF 657), monoclonal mouse anti-FLAG M2 (AF 488), or polyclonal rabbit anti-V5 (AF 647). (B) H1299 cells were infected with λ2-FLAG, λ2-V5, or both viruses at an MOI of 5, fixed at 12 hpi, and immunostained with monoclonal mouse anti-FLAG (AF 488, green) and polyclonal rabbit anti-V5 (AF 647, magenta) and imaged via immunofluorescence confocal microscopy. Created in BioRender. *Shmulevitz, M.* (2025) https://BioRender.com/3n918ke **(C)** Equivalent particles of T3D^PL (WT), λ2-FLAG, and λ2-V5 were used to infect L929 cells at 4°C for 1 hour. After 1 hour, cells were collected using cell stripper and fixed. Cells were immunostained with mouse monoclonal anti-σ3 (10G10, AF 647) and the amount of virus bound was quantified by flow cytometry (MFI). Data is representative of three independent experiments. (D-F) H1299 or L929 cells were infected with WT, λ2-FLAG, or λ2-V5 at a MOI of 5. After one hour of infection, virus-containing media was replaced with complete media with or without 100 µg/mL of cycloheximide (CHX). Cell lysates were collected every three hours from 9-18hpi for either SDS-PAGE or plaque assay. (D) Quantifications of reovirus proteins by SDS-PAGE and Western blot analysis of cell lysates with polyclonal rabbit sera directed against reovirus, and monoclonal mouse anti-β-actin. Data are representative of three independent experiments. (E) Representative Western blots from which quantifications in (D) were drawn. (F) Viral titers as measured by plaque assay, and statistical analysis is reported as unpaired t-test between the mean of each column. ****p < 0.0001, ***p < 0.001, **p < 0.05, ns > 0.05. Data are representative of

four independent experiments. All data is plotted as mean +/- 95% CI. Statistical analysis is reported as ordinary one-way ANOVA between the mean of each column (binding assay, **C**) or timepoint (D). ****$p < 0.0001$, ***$p < 0.001$, **$p < 0.05$, ns $> 0.05$.

compartment? To explore this question, AF-labelled cores generated from λ2-FLAG or λ2-V5 viruses were transfected into H1299 cells, fixed at 7 hpt, and immunostained with α-FLAG antibodies to discriminate between input and *de novo* protein-containing compartments (Figs 4A, B and S5A–C). Among the *de novo* λ2-FLAG-containing factory population, just over 50% contained no input cores, suggesting that *de-novo* proteins establish new factories devoid of input cores altogether (Figs 4B "*de novo* FLAG alone", C, and S5D, E). Approximately 21% of *de-novo* λ2-FLAG associated with factories produced by the parent input core (AF-λ2-FLAG) (Figs 4B "*de novo* FLAG and input FLAG", C, and S5D, E), while only ~10% co-localized with their non-self input (AF-λ2-V5), and ~13% co-localized with factories positive for both input particles (Figs 4B "*de novo* FLAG and input FLAG and input V5", C and S5D, E).Through a complementary analysis, a plurality of input AF-λ2-V5-positive compartments did not contain *de novo* or input λ2-FLAG (~47%), where the remaining puncta (with *de novo* and input λ2-FLAG, with only *de novo* λ2-FLAG, or with only input λ2-FLAG) each made up 14–19% of the population (Figs 4D, S5F, G). Considering these results together, it appears that the majority of *de novo* core proteins (~56%) become directed to new factories devoid of input cores, and the majority of input-core-containing factories (~47%) do not contain *de novo* proteins produced by non-self cores. Nevertheless, some (~15–20%) factories produced by input cores do receive *de novo* proteins produced by non-self cores.

Although the majority of input-core-containing factories did not contain non-self cores or *de novo* proteins, an assessment of the relative volumes for each unique factory composition provided new insights into the expansion and mixing that occur between peripheral factories. The volumes of factories containing both input λ2-FLAG and input λ2-V5 cores were larger than those of factories with only one input core population (Fig 4E), suggesting mixing or merging of early single-input-core peripheral factories. In all cases, factory volumes were significantly larger when they also contained *de novo* core proteins produced by the reciprocal virus, with the largest volumes representing both input cores and non-self *de novo* core proteins. The analysis suggests that while input cores initially form a majority of independent peripheral factories with minimal *de novo* proteins from other input cores, larger, more-mature factories ultimately contain mixtures of input cores and *de novo* proteins (Fig 4F). Moreover, *de novo* proteins can also create new peripheral factories devoid of input cores, but such factories are smaller than those formed from input cores.

### As replication progresses, *de-novo* core proteins start to mix indiscriminately between independent compartments created by input cores

The focus on OC-negative peripheral factories (Fig 4) revealed that while the majority of core-only factories were composed of independent input cores and self-derived de novo proteins, some new core-only factories contained a mixture of different input cores and non-self de novo proteins. In addition, other factories were formed entirely by de novo core proteins, devoid of input cores. To understand how composition changes as factories transition from peripheral core-only to mid-cytoplasmic core-plus-OC compartments, we evaluated the distribution of *de novo* proteins from two independent sources (λ2-FLAG and λ2-V5 viruses) between core-only (OC-negative) peripheral factories and OC-containing intermediate compartments. Instead of using labelled input cores, a *bona fide* virus infection cycle was used, where H1299 cells were co-infected with λ2-FLAG and λ2-V5 viruses. Infected cells were fixed at 8 hpi, immunostained with α-FLAG and α-V5 antibodies to detect core λ2 proteins, as well as with OC σ3 protein-specific antibodies to discriminate between OC(-) and OC(+) factories (Figs 5A and S6). Following IF-CM analysis, there was an equivalent proportion of the OC(-) peripheral factories positive for *de novo* λ2-FLAG (FLAG+) only, *de novo* λ2-V5 (V5+) only, and both *de novo* λ2-FLAG and λ2-V5 (FLAG+/V5+) (Fig 5B). However, the factories that contained both *de novo* λ2-FLAG and λ2-V5 were significantly larger in volume relative to the de-novo λ2-FLAG or λ2-V5 only factories (Fig 5C), despite all three localizing at

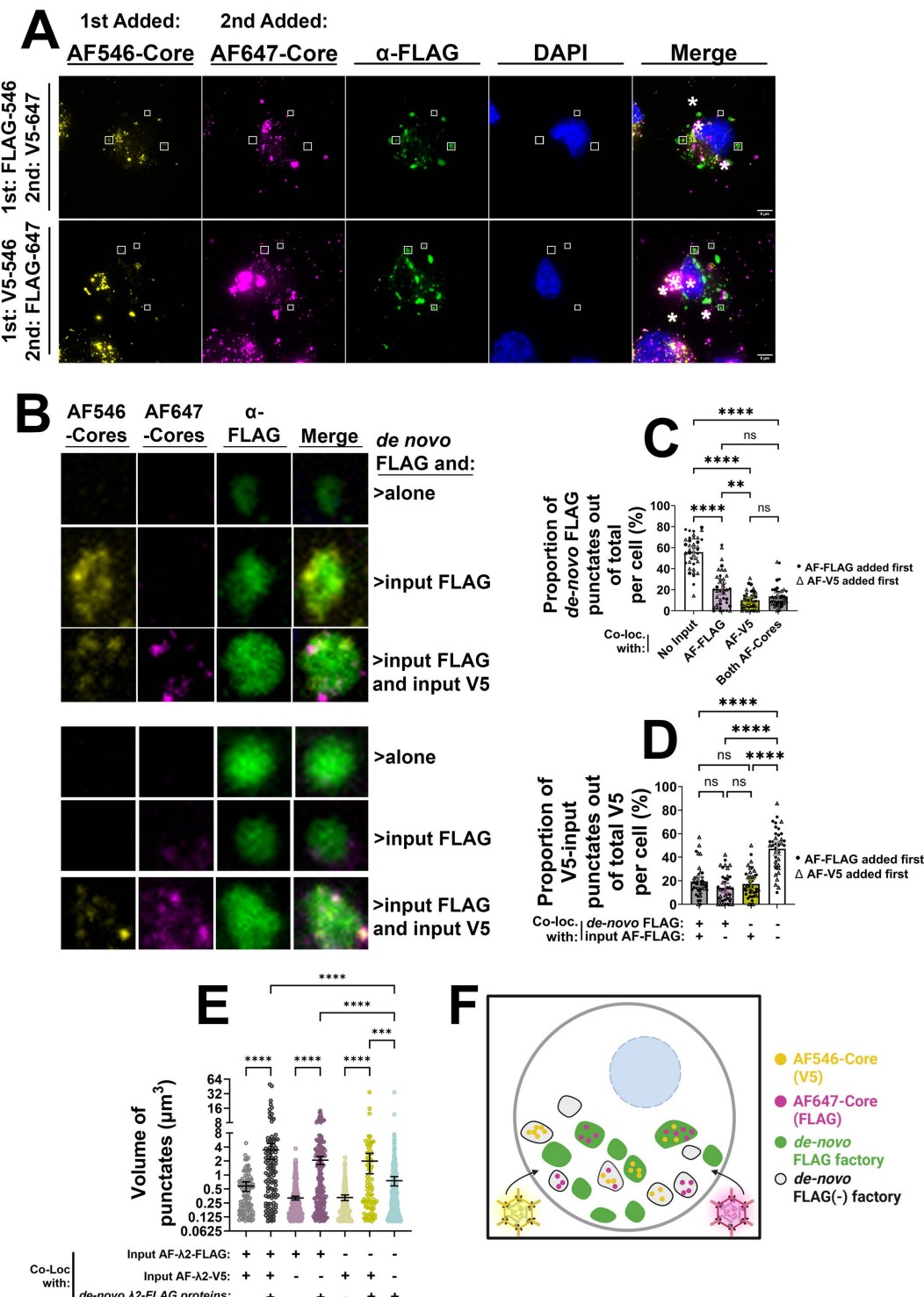

**Fig 4. In early peripheral factories, most of the independent input core-containing OC(-) factories do not acquire newly synthesized core proteins from external factories.** H1299 cells were first transfected with ~1000 AF-546 labelled FLAG- or V5-tagged cores per cell. 1 hour later, the cells were transfected with ~1000 particles per cell of AF-647 labelled FLAG- or V5-tagged cores (opposite tag to the first transfection). 7 hpt, cells were fixed and processed for immunofluorescence confocal microscopy. (A-B) Representative images of compressed Z-stacks. Cells were immunostained with monoclonal mouse anti-FLAG (AF 488, green) and stained with DAPI to visualize nuclei. White asterisks indicate aggregates formed by AF-cores. (B) Compartments of different identities (*de novo* FLAG alone, *de novo* FLAG with input FLAG, *de novo* FLAG with both input FLAG and input V5) were

selected (white boxes in (A) and artificially blown-up using Adobe Photoshop software. (C-D) The number of AF-FLAG cores, AF-V5 cores, and *de novo* λ2-FLAG punctates and the degree of colocalization between each were quantified. Circle data points represent data where AF-FLAG cores were transfected first, and triangles represent data where AF-V5 cores were added first. (C) The number of *de novo* λ2-FLAG (green) punctates were quantified and the proportion of punctates per cell lacking both AF-FLAG and AF-V5, containing only AF-FLAG, containing only AF-V5, or lacking both AF-cores were plotted. (D) The number of AF-V5 core punctates were quantified and the proportion of punctates co-localizing with both *de novo* and AF-FLAG, *de novo* FLAG only, AF-FLAG only, or neither *de novo* nor AF-FLAG were plotted. (E) The volume of punctates were quantified. Groups are categorized based on co-localization of input AF-FLAG cores, input AF-V5 cores, and/or *de novo* λ2-FLAG proteins. (F) Cartoon schematic, created in BioRender. *Shmulevitz, M.* (2025) https://BioRender.com/nqc59nl, summarizing the findings from A-E. All data is representative of four independent experiments with a minimum of five cells per condition analyzed. All data is plotted as mean +/- SD. Statistical analysis is reported as ordinary one-way ANOVA between the mean of each column. ****$p < 0.0001$, ***$p < 0.001$, **$p < 0.05$, ns > 0.05.

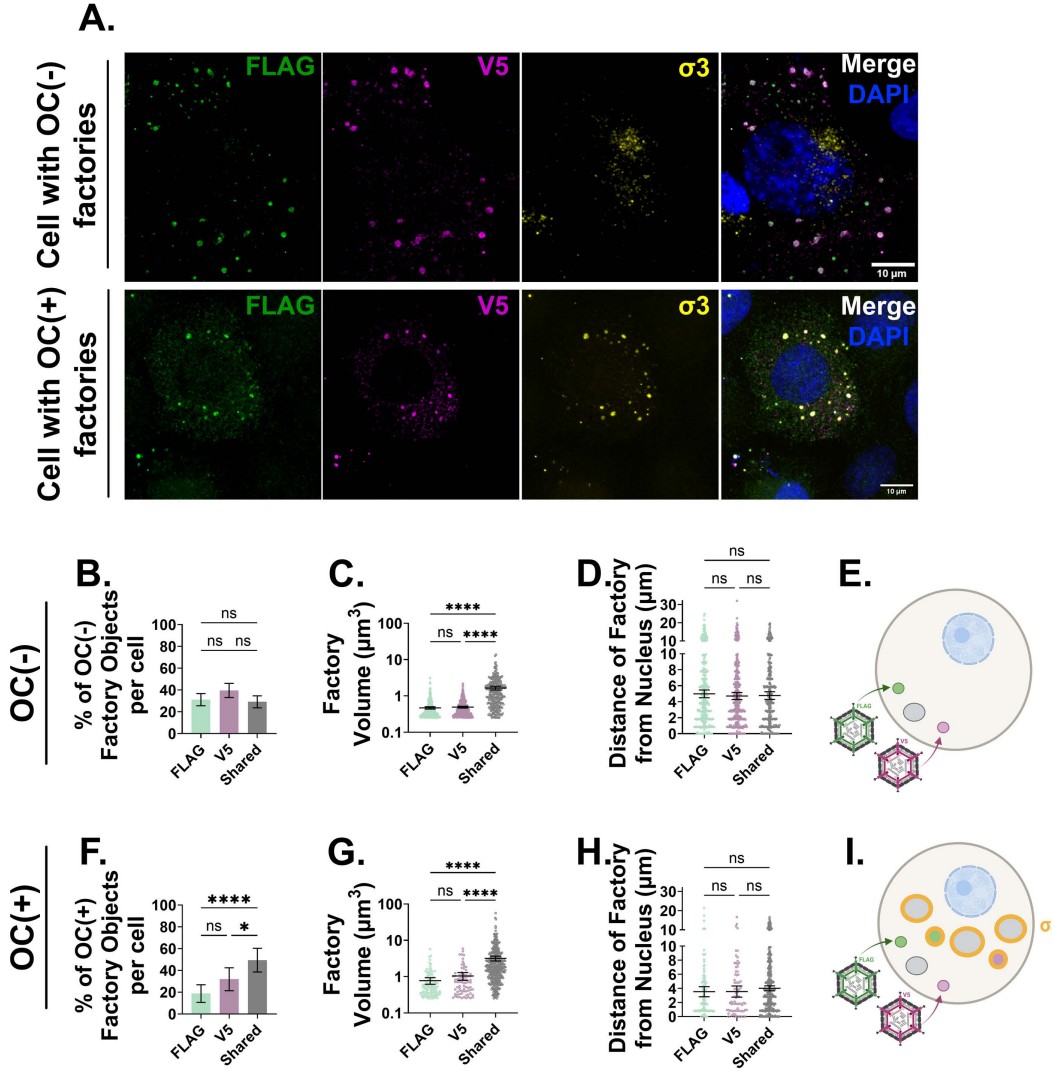

**Fig 5. *De novo* core proteins start to mix indiscriminately between independent compartments created by input cores.** H1299 cells were co-infected with λ2-FLAG and λ2-V5 at an MOI of 5 each. 8 hpi, cells were fixed and immunostained with monoclonal mouse anti-FLAG (AF 488, green), polyclonal rabbit anti-V5 (AF 647, magenta) and monoclonal mouse anti-σ3 directly conjugated to Alexa Fluor 594 (10C1, yellow) and stained with DAPI for nuclei visualization (blue). (A) Representative immunofluorescence confocal images. (B-D) Quantifications of OC-negative factory populations. (B) The number of FLAG(+), V5(+), or FLAG and V5(+) (Shared) OC(-) factory objects per cell graphed as a percentage of the total outercapsid-negative factory objects per cell. The (C) volume and (D) edge-to-edge distance from the nucleus of FLAG(+), V5(+), or shared OC(-) factory objects. (E) Cartoon model of the OC(-) factory populations observed at 8hpi. Created in BioRender. *Shmulevitz, M.* (2025) https://BioRender.com/olzwmfp. (F-H)

Quantifications of OC(+) factory populations. (F) The number of FLAG(+), V5(+), or shared OC(+) factory objects per cell graphed as a percentage of the total OC(+) factory objects per cell. The (G) volume and (H) edge-to-edge distance from the nucleus of FLAG(+), V5(+), or shared OC(+) factory objects. (I) Cartoon model of the OC(-) and (+) factory populations observed at 8 hpi. Created in BioRender. *Shmulevitz, M.* (2025) https://BioRender.com/7blhjp7. All data is representative of three independent experiments with a minimum of 5 cells analyzed per experiment. Data is plotted as mean +/- 95% CI. Statistical analysis is reported as ordinary one-way ANOVA between the mean of each column. ****$p < 0.0001$, ***$p < 0.001$, **$p < 0.05$, ns > 0.05.

approximately the same peripheral region of the cell (Fig 5D, E). The larger size of OC(-) peripheral factories containing a mixture of λ2-FLAG and λ2-V5 *de novo* proteins may represent new factories formed by both *de novo* proteins and/or independent factories that merge into larger shared ones.

With respect to the OC(+) population, which represented factories further along in the replication cycle and containing *de novo* OC proteins (σ3), there were equivalent proportions of smaller, independent FLAG+ and V5+ factories, but a significant increase in the proportion of FLAG+/V5+ factories, which were larger in volume compared to only FLAG+ or only V5+ factories (Fig 5F, G). All OC(+) factories were located at similar distances from the nucleus, though they were closer to the nucleus than the OC(-) factories (Fig 5H, I). These observations suggest that as early as 8 hpi, when core amplification is exponential, *de novo* core proteins from different viral sources co-localize within the same factories, with approximately one-third of OC(-) core-only factories and one-half of OC(+) factories containing shared *de novo* proteins. This co-localization of proteins from different viral sources raises the possibility that core amplification factories could contribute proteins with distinct mutations or functions, although future studies would be needed to directly establish the extent of functional complementation.

Altogether, Figs 1–5 reveal the dynamic formation of peripheral core-only factories and transition to core-plus-OC mid-cytoplasmic factories as follows: Input core particles first seed independent initial OC(-) factories in the cell periphery, with the majority retaining only self-derived *de novo* proteins. As infection progresses, these input(+) factories move into intermediate regions where they acquire OC proteins. During this process, the originally independent factories are more likely to merge and/or acquire *de novo*-produced products from non-self origins, resulting in the "shared" factory phenotype. While the first wave of core-only factories move into intermediate regions, new *de novo* OC(-) factories that lack input particles are simultaneously seeded. During this second wave of replication, products mix indiscriminately, forming an equivalent amount of independent and shared *de novo* OC(-) factories. By late infection, input particles accumulate in OC(+) perinuclear factories of shared phenotype, where all components have merged, while new de novo OC(-) factories continue forming in the periphery during ongoing rounds of replication.

### Microtubule polymerization does not affect OC(-) peripheral factories and their transition to OC(+) mid-cytoplasmic factories but is required for transition of whole assembled virions into perinuclear depots

The transition of independent input cores from peripheral core-only factories to mid-cytoplasmic OC+ factories, their later transitions into perinuclear condensates, and the sharing of *de novo* core proteins suggest a movement of factories and proteins that could be mediated by passive mechanisms such as diffusion along concentration gradients, or active mechanisms such as actin-based and/or microtubule-dependent transport [55]. Previous studies found that an intact microtubule (MT) network and motor protein dynein are necessary for factory accumulation at perinuclear areas [56,57]. However, these studies focused on late time points when most core amplification and progression into perinuclear factories would have already occurred and were conducted prior to discovery of distinct core-only versus core+OC factory types. Nocodazole treatment was also previously suggested to reduce genome packaging efficiency [58], however recent cryoET studies [59] suggest that electron microscopy structures previously thought to represent genome-devoid whole particles may sometimes represent shells formed by outercapsid proteins rather than empty whole particles. Accordingly, to explore the earlier stages of factory dynamics during reovirus replication, the role of MT was revisited at time points that capture exponential core amplification and all three compartment types: core-only peripheral factories; mid-cytoplasmic core+

OC+ factories, which are largely devoid of fully assembled viruses but adjacent to OC-decorated lipid droplets; and perinuclear compartments that house fully assembled virions.

To test the role of MTs during factory transitions, wild-type T3D$^{PL}$ reovirus-infected cells were treated with the MT depolymerizing drug nocodazole at 1 hpi for the duration of infection. Nocodazole was added post-virus adsorption to avoid potential effects of MT disruption on reovirus endocytosis [50]. Cells were fixed at 8 and 12 hpi, followed by IF-CM to monitor protein and factory distribution and composition (Figs 6 and S7). Importantly, β-Tubulin staining appeared filamentous under control DMSO treatment but diffuse and granular following nocodazole treatment, indicating that MT integrity was successfully disrupted by nocodazole treatment (Figs 6A, 100x β-Tubulin panels in magenta, S7A,B Fig). When IF-CM with polyclonal core protein–specific antibodies and OC σ3 was used to distinguish core-only from core+OC factories, there was no difference between DMSO and nocodazole treatment in the percentage of OC(-) and OC(+) factories (Fig 6B), nor in the volumes of either OC(−) or OC(+) factories (Fig 6C). These results suggest that MT networks are not required for the establishment or growth of core-only factories, nor for their transition into mid-cytoplasmic core-plus-OC factories.

As was previously observed by others at late timepoints of infection [56], there was a lack of large interconnected perinuclear factories adjacent to the nucleus following nocodazole treatment compared with DMSO (Figs 6A [σ3 and core staining in bottom panels of both 100x and 60x] and S7C). Quantitatively, OC(-) factories remained significantly further from the nucleus at both 8 and 12 hpi, and OC(+) factories at 12 hpi following nocodazole treatment (Fig 6D). Moreover, because previous findings have implicated an association between OC proteins and LDs that may impact assembly [49,60], the number of LDs and the proportion associated with OC proteins following nocodazole treatment were also assessed. Quantitative IF-CM revealed that neither metric was altered following nocodazole treatment at either 8 or 12 hpi (Fig 6E, F). In addition to associating with OC proteins, a previous study observed that LDs can congregate closer to the nucleus and each other following reovirus infection [49]. Both of these effects were reduced following nocodazole treatment; LDs were found farther from the nucleus (Fig 6G) and farther away from each other than in the DMSO control (Fig 6H). Altogether, the early events of reovirus amplification, including the formation of core-only factories, core-plus-OC factories, and LD-associated OC proteins, can occur and transition to mid-cytoplasmic compartments in an MT-independent manner. However, in the absence of an intact MT network, factory progression is stalled at the mid-cytoplasmic OC(+) intermediate phase and fails to produce the large perinuclear depots of fully assembled virus particles.

Results presented previously demonstrated that input core particles transition from predominantly OC(-) factories in the cell periphery to OC(+) factories in intermediate and perinuclear areas as infection progresses (Fig 1). When MTs were disrupted, OC(+) factory progression became stalled at the mid-cytoplasmic location with circular-factory morphology, rather than progress to the amorphic nuclear-proximal location (Fig 6). The OC(-) factories were still present in the periphery. To determine if the dynamics of input core particles could be altered following nocodazole treatment, or in other words, if the localization of input particles might be MT-dependent, H1299 cells were transfected with AF546-cores, fixed at 12 hpt, and immunostained with antibodies directed against core proteins and OC σ3 (Fig 7). 1 hpt, transfected cells were treated with the MT depolymerizing drug nocodazole at 1 hpi for the duration of infection, or with DMSO as a control (Figs 7A, B, and S8A, B). Nocodazole-mediated MT disruption did not affect the proportion of factories associated with input AF-cores (Fig 7C) nor the proportion of AF-cores associated with OC(+) factories (Fig 7D) at 12 hpt. Moreover, single pinpoint foci of AF-core fluorescence could be easily visualized and counted within factories, with average diameter of 464nm and high circularity. The pinpoints were unlike the aggregates demarked by asterisks, which were much larger, were poorly circular, and were never associated with factories. Although the measured diameter of each pinpoint exceeds the size of a single virus particle (60–80nm), objects smaller than the ~200nm diffraction limit of confocal microscopy often appear blurred, causing their fluorescence signals to be larger than their true size. Consequently, it was not possible to definitively determine whether each pinpoint focus represented a single core or multiple cores clustered together. Nonetheless, the number of pinpoints per factory ranged from 0 to over 50. Counting these pinpoints revealed that factories

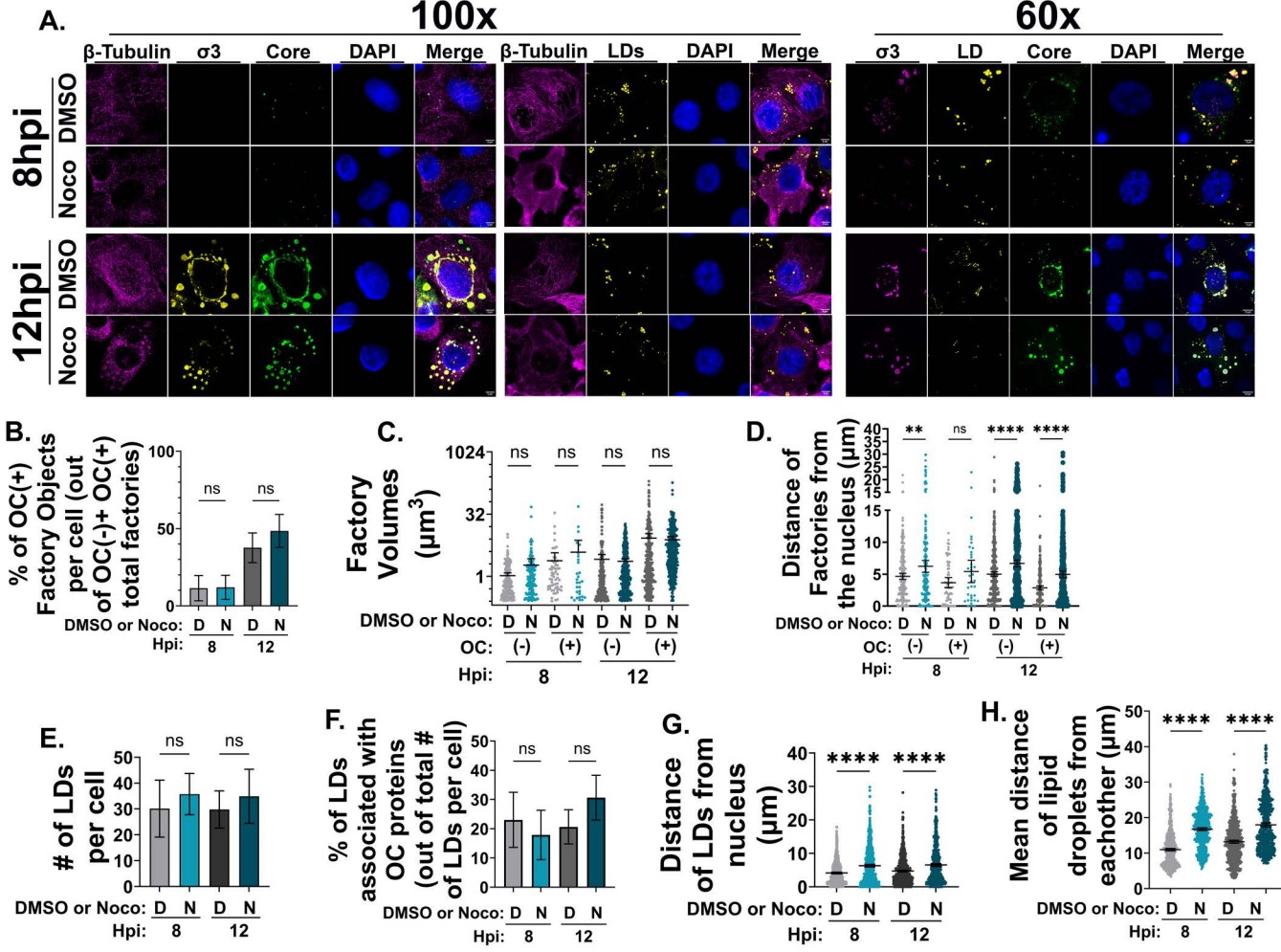

**Fig 6. Microtubule polymerization does not affect OC(-) peripheral factories and their transition to OC(+) mid-cytoplasmic factories but is required for transition of whole assembled virions into perinuclear depots.** H1299 cells were infected with T3D$^{PL}$ MOI 3. 1 hpi, complete media containing 10μM nocodazole, or an equivalent volume of DMSO (control), was added to the cells. 8 or 12 hpi, cells were fixed and immunostained for immunofluorescence confocal microscopy imaging. (A) Representative images of cells stained with: monoclonal mouse anti-tubulin (12G10, AF 647, magenta), BODIPY 493/503 for lipid droplet visualization (LDs, yellow), DAPI for nuclei visualization (blue), monoclonal mouse anti-σ3 (10C1, directly conjugated to AF 594 in the middle panel (yellow), or 10G10 followed by AF 647 conjugated secondary antibodies in the right panel (magenta)), and/or polyclonal rabbit antibodies raised against reovirus cores (AF 488, green). Images were captured at 100x and 60x magnification. Images were created from Z-stacks and are representative of a minimum of five images captured for each condition from three or four independent experiments. (B) The number of factories positive for OC staining were quantified and graphed as a percentage of total factories per cell. (C) The factory object volumes and (D) edge-to-edge distances from the nucleus were quantified. (F) The total number of lipid droplets per cell were quantified. (F) The number of lipid droplets associated with OC proteins per cell were graphed as a percentage of the total number of lipid droplets. (G) The edge-to-edge distance to the nucleus of each lipid droplet was graphed. (H) The mean edge-to-edge distance between each lipid droplet per cell was graphed. All data is representative of 3 independent experiments and plotted as mean +/- 95% CI. Statistical analysis is reported one-way ANOVA between the mean of each column. ****$p < 0.0001$, ***$p < 0.001$, **$p < 0.05$, ns > 0.05.

with only a few input cores behaved similarly to those with many input cores, suggesting that the number of input cores does not alter the phenotype of the factory. Importantly, nocodazole treatment did not significantly alter the distribution of input core number per factory. These data suggest that the distribution of AF-cores across factories, including those containing OC proteins, is not mediated by MTs.

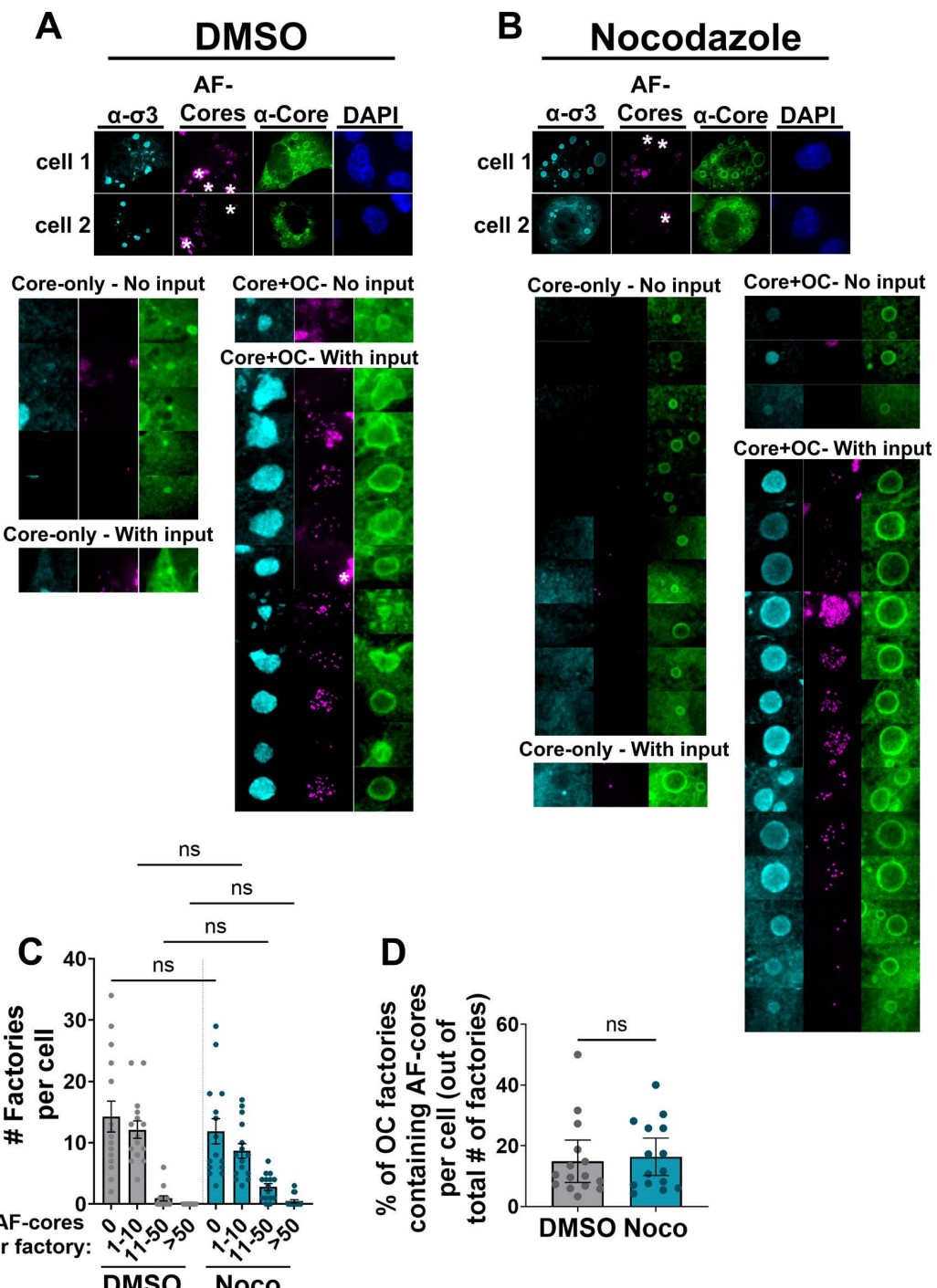

**Fig 7. The localization of input AF-core particles is not microtubule dependent.** H1299 were transfected with ~1000 AF-546 reovirus core particles per cell (magenta). 1 hpt, cells were treated with (A) DMSO or (B) 10μM nocodazole. 12 hpt, cells were fixed and immunostained with monoclonal mouse antibodies directed against σ3 (10G10, α-σ3, AF 647, cyan) polyclonal rabbit antibodies raised against reovirus cores (α-core, AF 488, green) and DAPI was used to stain nuclei. All images are of compressed Z-stacks captured by IF-CM. White asterisks indicate aggregates formed by AF-cores. Factory objects within the cell were artificially blown-up using Adobe Photoshop software to showcase regions containing *de novo* cores only – no input particles, *de novo* cores with input, core and OC with no input, and core and OC with input. (C) The number of factories per cell, and the number of input AF-core particles within the factories, were quantified from 3D unedited images using Volocity software. Statistical analysis is reported one-way ANOVA between the mean of each column. ****$p < 0.0001$, ***$p < 0.001$, **$p < 0.05$, ns $> 0.05$. (D) The percentage of OC factories that contained input AF-cores

out of the total number of factories per cell were quantified. Statistical analysis is reported as unpaired t-test between the means of the columns. ****$p < 0.0001$, ***$p < 0.001$, **$p < 0.05$, ns > 0.05. All data is plotted as mean +/- 95% CI. All data is representative of a minimum of five cells per three independent experiments.

Numerous studies have suggested that while some laboratory strains of reovirus such as T3D$^{PL}$ and T1L produce similar factory morphologies, these differ from the factory morphology of the T3D$^N$ laboratory strain. The differences in factory morphologies were previously associated with the amino acid at position 208 of the MT-associating μ2 protein [57]. It was important, therefore, to establish if the factory characteristics discovered in our studies using T3D$^{PL}$ are representative of reoviruses more generally. Environmentally derived reoviruses that have not undergone lab culture adaptation provide a strong reflection of the reovirus quasispecies as a whole. The phenotypes observed for T3D$^{PL}$ following nocodazole treatment (Fig 6) were consistent among environmentally derived reoviruses (T1E1, T2E1, T2E2), as well as laboratory strain T1L (S9A Fig). Moreover, with the exception of T3D$^N$, the remaining viruses all share the same amino acid at position 208 of μ2 (S9B Fig). Accordingly, the role MT networks play in factory transitions is not limited to only T3D$^{PL}$ or T1L reoviruses but applies to naturally derived reoviruses as well.

### Microtubule polymerization in part mediates the core to whole virus transition

Although nocodazole treatment inhibited the accumulation of large perinuclear factories, OC(+) factories still formed in intermediate areas. Previous studies using nocodazole focused on late time points of infection and reported similar results [57]. Studies assessing the effects of nocodazole on reovirus titers, however, produced conflicting results. While one study found no significant differences in titers at 24 hpi following nocodazole treatment beginning at 6 hpi with either T1L or T3D$^N$ [61], another study using the same experimental approach found an approximately 50% reduction in titers at 24 hpi using the T1L strain. No significant reduction was observed with the T3D strain, although it was not explicitly stated which T3D strain was used [58]. Hence, to investigate whether nocodazole treatment impacts T3D$^{PL}$ progeny titers over the course of replication, T3D$^{PL}$-infected H1299 cells, with or without nocodazole treatment, were subjected to single-step growth kinetic analysis. There was a ~10-fold decrease in viral progeny titers under nocodazole treatment beginning at 12 hpi (Fig 8A). To determine if the diminished progeny production could have been due to a decrease in *de novo* protein production following MT disruption, duplicate wells were subjected to SDS-PAGE and quantitative Western blot for assessment of reovirus protein levels (Fig 8B). An ~2-fold decrease in core protein production at 12 hpi following nocodazole treatment was observed (Fig 8C), with a similar decrease in OC protein expression between 12–18 hpi (Fig 8D). However, a 2-fold decrease in protein production mid-to-late infection was unlikely to fully account for a 10-fold decrease in progeny virus generation.

The effects of nocodazole treatment were explored further by electron microscopy as a direct but qualitative assessment (S10 Fig). Recent cryo-electron tomography studies have identified multiple structures within infected cells, including virus cores, intact virions containing dsRNA genomes, empty virions lacking genomes, and newly discovered outer capsid (OC) protein shells [59]. Both dsRNA-containing virions and OC shells were observed within peripheral and mid-cytoplasmic factories, in both nocodazole-treated and untreated samples. The number of outer capsid shells varied significantly among different cells and even among factories within the same cell, making precise quantification from single-plane EM images challenging. Nevertheless, qualitatively, there was no striking difference in the overall abundance of outer capsid shells between treated and untreated cells. Minimal numbers of empty, genome-devoid virions were observed in either condition, suggesting that aberrant genome packaging is unlikely to account for the approximately 10-fold difference in viral titers. The most notable difference observed was that in untreated cells, microtubules appeared to infiltrate mid-cytoplasmic factories. Fully assembled virions accumulated with increasingly greater density within microtubule-rich domains, giving the impression that microtubules were releasing, guiding and/or extracting mature particles from the factories and redistributing them into increasingly tighter fully assembled virion storage depots.

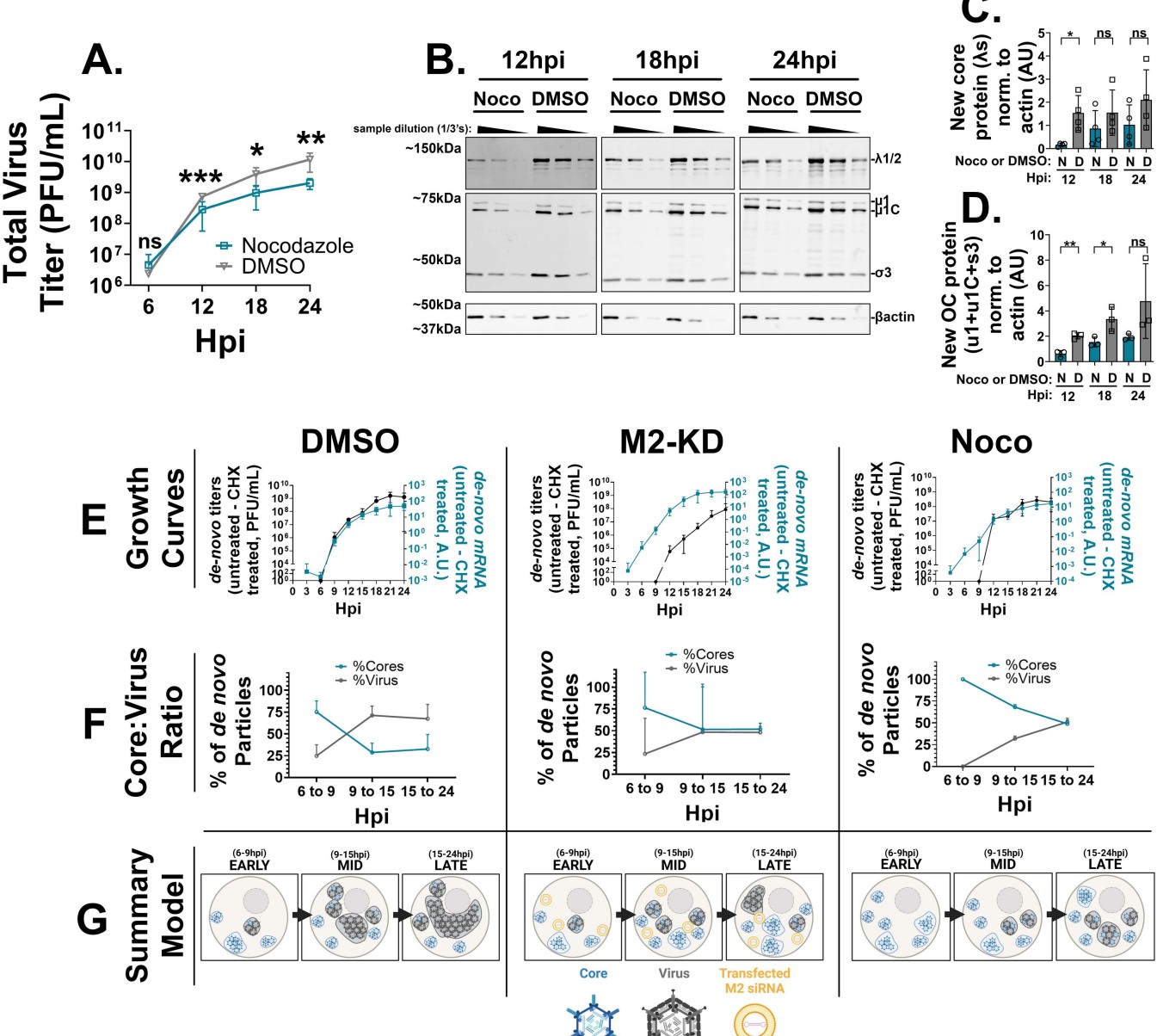

**Fig 8. Nocodazole-mediated microtubule disruption delays progeny virus full assembly.** (A-D) H1299 cells were infected with T3D^PL at an MOI of 3. 1 hpi, media containing either 10µM nocodazole or an equivalent volume of DMSO (control) were added to the cells. (A-B) Beginning at 6 hpi, cell lysates were collected every 6 hours for 24 hours. (A) The total virus titer from media plus cell lysates were determined. Titer data is plotted as mean +/- SD. Statistical analysis is reported one-way ANOVA between the mean of each timepoint. ****$p < 0.0001$, ***$p < 0.001$, **$p < 0.05$, ns $> 0.05$. (B-D) Beginning at 12 hpi, cell lysates were collected every 6 hours for 30 hours. Lysates were subject to SDS-PAGE and Western blot analysis using polyclonal rabbit sera directed against reovirus and monoclonal mouse anti-β-actin. (B) Representative blots of dilution series of each sample for accurate quantification. Western blot was performed with antibodies directed against purified core particles (α-core, top), whole virus particles (α-reo, middle) and β-actin (bottom) Quantification of (C) core proteins(λ1/λ2) normalized to β-actin and **(D)** outercapsid proteins (μ1+μ1C+σ3) normalized to β-actin. Western blot quantifications are plotted as mean +/- SD. Data from **(A)** is representative of four independent experiments, data from (B-D) are representative of three independent experiments and statistical analysis is reported as multiple unpaired t-tests between the mean of each timepoint and/or column ****$p < 0.0001$, ***$p < 0.001$, **$p < 0.05$, ns $> 0.05$. (E-F) H1299 cells were first transfected with siRNAs against the M2 gene, or a non-targeting control (in DMSO alone and Noco columns). Cells were infected with T3D^PL at an MOI of 3. 1 hpi, cells were treated with 10µM nocodazole or DMSO in the presence of absence of 100µg/mL cycloheximide. Every three hours for 24 hours, both RNA and cell lysates were collected to measure *de novo* core and virus generation, respectively. (E) The single-step growth kinetics of whole viruses (black) and cores (blue). (F) The percentage of core particles

(blue) and whole virus particles (grey) out of the total particles grouped over the timeframes. (G) Cartoon schematics summarizing the findings shown in (E) and (F)created in BioRender. *Shmulevitz, M.* (2025) https://BioRender.com/ocj82yz for DMSO summary, for the M2KD summary: *Shmulevitz, M.* (2025) https://BioRender.com/br68zkb, for the NT summary: *Shmulevitz, M.* (2025) https://BioRender.com/xrumjpx, and for the bottom icons: *Shmulevitz, M.* (2025) https://BioRender.com/ydy4k1q. Data for (E) and (F) is plotted as mean +/- SD and represents three independent experiments.

Given that the major effect of disrupting MT networks was cessation of perinuclear factories that normally represent areas of fully assembled virus deposition, we wondered if the reduced titers in nocodazole-treated cells might reflect a change in the transition of core particles to fully assembled particles. To determine the impact of nocodazole on particle assembly, the kinetics of core and whole virion amplification over the first round of infection were assessed. H1299 cells were infected with reovirus in the presence or absence of nocodazole, and every 3 hours for 24 hours, RNA and whole cell lysates were collected to measure core and virus production, respectively. Since cores synthesize RNAs, RT-qPCR of reoviral RNAs was used as a proxy for core amplification. Whole virus levels were simultaneously measured by plaque titration. To differentiate RNA and whole virions synthesized linearly from input particles as opposed to logarithmic amplification from *de-novo* synthesized particles, cycloheximide was used to thwart secondary rounds of viral amplification by inhibiting translation. As a positive control for delayed OC assembly onto cores, cells were treated with siRNAs to knockdown the M2 gene, that encodes the OC µ1 protein, prior to infection in the absence of nocodazole. The single-step kinetics of core-derived RNAs and whole infectious virions were logarithmic during normal infection (Fig 8E, left (DMSO)). Core RNA amplification was not disrupted by nocodazole treatment. However, compared to DMSO-treated cells, there was a delay in whole virus production relative to core amplification in both M2-siRNA– and nocodazole-treated cells (Fig 8E, middle (M2-KD) and right (Noco), respectively). When plotted as the ratio of core-to-whole-virus production at early (6–9 hpi), mid (9–15 hpi), and late (15–24 hpi) stages of reovirus replication (Fig 8F), it was evident that whole virus assembly was not only delayed relative to core amplification under MT disruption, but that fewer whole viruses were ultimately produced. These results suggest that MTs are necessary not only for depositing whole viruses from mid-cytoplasmic core-plus-OC factories into perinuclear compartments (Fig 8G), but also for ensuring the timely and efficient transition of cores into whole viruses.

## Discussion

Until recently, replication and assembly of segmented dsRNA mammalian orthoreovirus was all thought to occur within a single, undifferentiated type of factory. However, the development of core-specific and outercapsid-specific antibodies has revealed a more complex, spatially organized process [49]. Core amplification is segregated in peripheral core-only factories, while OC proteins localize separately on host LDs. Mixing of core and OC components takes place at intermediate factories, and fully assembled virions are ultimately deposited in perinuclear arrays [49]. The current study aimed to: (1) determine the role of input cores in the formation of peripheral core-only factories, (2) assess whether core-only factories directly transition into intermediate core+OC factories, (3) establish whether *de novo* viral core proteins mix in peripheral core-only factories and/or intermediate core+OC factories, (4) define the role(s) of microtubules (MTs) in formation and transitions between factories, and (5) map the temporal dynamics of such processes. Each experiment was designed to distinguish between different possible scenarios. For example, factories could have become established at locations distant from input cores using products of input cores, or factories could have instead formed directly surrounding the input cores themselves; the data now supports the latter scenario. Similarly, the OC protein-containing mid-cytoplasmic factories could have been created by movement and merging of the peripheral factories or instead the peripheral and mid-cytoplasmic factories could co-exist independently akin to how cellular compartments can remain independent while sharing resources; the data now indicates that the former scenario is true. The movement of peripheral factories towards mid-cytoplasmic factories may have been tubulin-dependent similar to late stages of perinuclear factory formation, but instead were herein found to be tubulin-independent. Additionally, factories formed from two different input

cores—distinguished by unique λ2 tags—could have immediately mixed their core proteins, implying a broad distribution of viral proteins from a distal source; however, our findings reveal that non-self core proteins are not equally distributed among self and non-self factories, supporting the idea of a local source of viral proteins. Unexpectedly, we also observe that new core-only replication factories are continuously seeded at the cell periphery throughout infection, representing an anomalous or surprising deviation from the typical inward movement pattern. Overall, by systematically testing these scenarios, the data allow us to propose a working model of reovirus replication—based on the most consistent interpretation of our findings—that describes the process as follows: (1) peripheral factories initially form around input cores (Fig 1) and contain predominantly self-cores and their corresponding *de novo* core proteins (Figs 2, 4, and 5A, E), (2) OC proteins are segregated away from core-only factories and deposited onto host LDs (Fig 6A, E–H) [49,60], (3) peripheral core-only factories transition into intermediate core+OC factories in a (MT)-independent mechanism (Figs 6A–D and 7), (4) new peripheral core-only factories can form without input cores and contain a mixture predominantly consisting of *de novo* core proteins from different input cores (Fig 4), (5) these *de novo* peripheral core-only factories also migrate into intermediate areas and mix with OC factories in a MT-independent process (Fig 6), and (6) cores within intermediate factories are coated with OC proteins and then deposited into the perinuclear compartment in a MT-dependent process (Figs 6 and 8). Overall, this study elucidates the intricate spatiotemporal dynamics of reovirus replication and assembly, highlighting distinct pathways and interactions between core and OC components essential to the productive infection cycle (Fig 9).

To unravel the temporal and spatial dynamics of reovirus replication, including the visualization of both input cores and *de novo* core proteins, we first developed two crucial methodologies. First, to visualize input cores, we produced cores from whole virions *in vitro* using standard proteolysis techniques [62,63], conjugated these cores to Alexa Fluor dyes (AF), and transfected the AF-cores into cells (Figs 1, S1–S5, S7, and S8 Figs). Second, to visualize *de novo* core proteins originating from distinct cores, we examined eight regions of the core λ2 protein and identified one site amenable to epitope tag insertion without abrogating assembly into transcriptionally active progeny cores (Fig 3 and S4). These methods can now be employed to further elucidate additional aspects of the spatiotemporal replication and assembly process. For instance, since input cores initiate the formation of the first core-only peripheral factories and subsequently transition into intermediate factories (Figs 1 and S2), future studies could use live-cell imaging techniques to track movement, merging, and transition of core-only and intermediate factories, as well as OC-containing LDs, over time. However, implementing such live imaging approaches will require substantial optimization to manage challenges such as photobleaching and signal degradation. Our inability to distinguish if pinpoint foci from AF-labelled input cores (e.g., Fig 7) represent a single input core versus several cores in close proximity, could be overcome in the future by using fluorescence microscopy techniques with constantly-improving diffraction limits and more accurate measurements of small structures. In addition to the better development of live-imaging approaches, future intel could be garnered through the use of aliphatic diols [64–66]. These compounds allow for the reversible dispersal of replication factories, and as such, may permit the tracking of redistributed factory components such as individual proteins or AF-cores. Furthermore, future studies can expand on these methodologies to monitor the localization of reovirus RNAs. Such efforts could reveal whether RNAs, like *de novo* core proteins, mix during the formation of the second round of core-only peripheral factories.

These results provide a clearer understanding of the mechanisms involved in core amplification during reovirus replication. While it was previously known that input cores could associate with µNS [36,67], our findings specifically show that these input cores initiate the formation of the first set of peripheral core-only factories (Figs 1 and S2). This suggests that RNA synthesis, expression of the factory-forming proteins µNS and σNS, and production of *de novo* core proteins from input cores likely occurs and condenses locally. The greatest support for this local condensation theory is the observation that when two differentially labeled cores were introduced into cells producing distinctively tagged λ2 core proteins, there was minimal mixing between the peripheral core-only factories seeded by different input core populations (Figs 2, S3–S5); if viral proteins were synthesized elsewhere and recruited to the input cores, then majority of input core factories should contain non-self proteins. Another key discovery is that peripheral core-only factories transition into intermediate core+OC

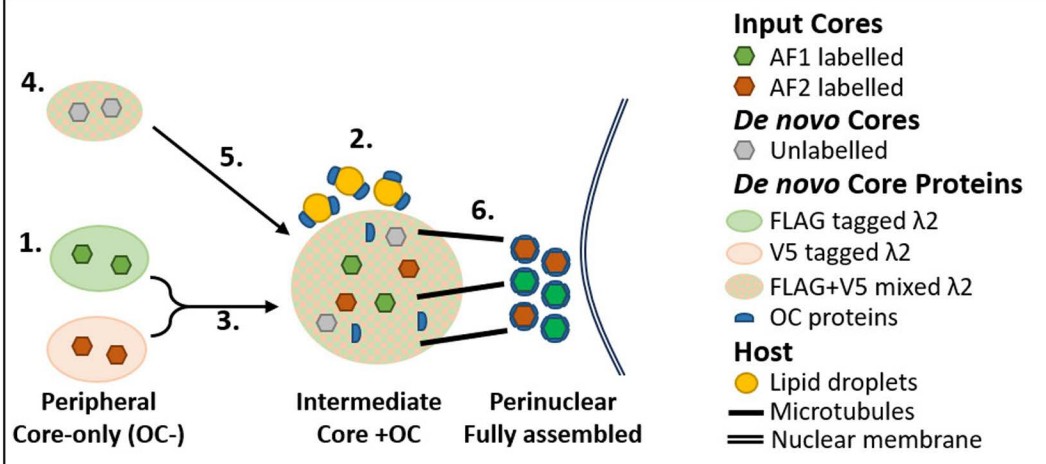

**Input Cores**
- 🟢 AF1 labelled
- 🔴 AF2 labelled

***De novo* Cores**
- ⬡ Unlabelled

***De novo* Core Proteins**
- 🟢 FLAG tagged λ2
- 🔴 V5 tagged λ2
- 🟤 FLAG+V5 mixed λ2
- 🔵 OC proteins

**Host**
- 🟡 Lipid droplets
- ━ Microtubules
- ═ Nuclear membrane

**Peripheral Core-only (OC-)**

**Intermediate Core +OC**

**Perinuclear Fully assembled**

1. Peripheral factories form around input cores. Such factories predominantly contain self cores and self *de novo* core proteins
2. Outercapsid (OC) proteins are segregated away from core-only factories and deposit on host lipid droplets
3. The peripheral core-only factories move into intermediate core+OC factories in a microtubule (MT)-independent manner
4. New peripheral core-only factories form devoid of input cores that contain predominantly a mixture of *de novo* proteins from different input cores.
5. *De novo* peripheral core factories devoid of input cores then migrate into intermediate areas and mix with core +OC factories in a MT-independent process.
6. Cores in the intermediate factory are coated with OC proteins and then deposited into the perinuclear compartment in a MT-dependent manner

**Fig 9. OC(-) factory dynamics proceed independent of microtubules, but intermediate OC(+) factories require intact MTs for transition into perinuclear areas. Cartoon diagram summarizing findings from Figs 1 through 8, modelling dynamics of two input AF-cores (AF647-FLAG and AF546V5) versus *de novo* produced λ2-FLAG and -V5 tagged products and MT interactions.** (1) During early infection, input cores form predominately independent OC(-) factories in the periphery of the cell. (2) Simultaneously, OC proteins are segregated from core-only factories and accumulate on LDs. (3) As input core-factories move from peripheral to intermediate regions during mid-infection in a MT-independent manner, they can acquire OC proteins and are more likely to merge together and/or acquire products from non-self factories resulting in a larger proportion of shared factories among the OC(+) factories. (4) At the same time, new core-only factories emerge in the periphery and contain *de novo* proteins from different input cores. (5) These newly produced factories then also move into intermediate spaces and mix with core and OC factories in an MT-independent process. (6) By late infection, OC proteins assemble onto input cores, and they are then deposited in perinuclear factories in a MT dependent manner.

factories as they move inward (Figs 1, S2, S6–S8), contrary to the alternative hypothesis that core-only factories remain at the periphery and *de novo* produced intermediate factories from rogue viral RNAs and proteins. By tracking fluorescently labeled core particles, we were able to distinguish between these hypotheses, clearly demonstrating that a directed transition does occur. Interestingly, while first-generation peripheral core-only factories migrate towards the center of the cell, new core-only factories consistently emerge at the periphery (Figs 1 and S2). These second-generation factories are devoid of input cores and contain a mixture of *de novo* core proteins derived from various input cores (Figs 4, S5 and S6). This continuous regeneration of peripheral factories likely serves as a strategic mechanism to sustain core amplification, while maintaining spatial separation from OC proteins and full virion assembly, a process that would otherwise terminate the transcriptional activity of newly formed active cores.

Previous studies found that reovirus μ2 interacts with MTs [57,68–71], and that nocodazole-mediated MT disruption leads to accumulation of large, rounded factories late in infection [56]. However, the role of MTs in early stages of

replication and the formation of distinct factory types has remained unclear. Our findings show that MTs are not essential for establishing core-only peripheral factories around input cores (Figs 6, S7 and S8), the formation of *de novo* peripheral core-only factories (Figs 7 and S8), or the transition of core-only factories into core+OC intermediate factories (Figs 6 and S7). This raises the question of what mechanisms facilitate the movement and maturation of core-only peripheral factories into core+OC intermediate factories. One possibility is that the movement and transition may rely on other components of the cytoskeletal network, such as actin filaments [72,73]. Actin could serve as a structural framework, facilitating directional movement and spatial organization of factories through dynamic remodeling and interactions with associated proteins. Alternatively, movement may occur via passive diffusion along concentration gradients, where the crowded perinuclear region may exert a "pull" that drives the movement of factories inward; however, we found no primary research that directly documents the movement of large complexes such as viral factories or organelles from the cell periphery to the perinuclear area by passive diffusion alone. Discriminating between these alternative pathways could offer valuable insights into the cellular infrastructure that supports viral replication and may reveal potential targets for antiviral strategies. Additionally, while it is logical that the first generation of core-only factories form at the periphery where input cores initiate replication, it is intriguing that the second generation of core-only factories, devoid of input cores, continue to emerge at the periphery. This suggests that assembly and condensation of *de novo* cores, along with factory-forming NS and core proteins, is persistently nucleated at the periphery. For other cellular processes, peripheral condensates can be formed by local translation at the cell periphery; for example, at neuronal synapses where RNA granules or translation-dependent condensates can form near synaptic terminals, in an MT-independent manner [74,75]. Likewise, stress responses like heat shock or oxidative stress can also nucleate new condensates near the plasma membrane using actin-based transport [76]. Understanding what drives peripheral nucleation of *de novo* core-only factories could provide significant insights into the spatial organization of viral replication within the host cell.

Consistent with previous findings [56], MT disruption caused accumulation of large, rounded factories in intermediate areas instead of perinuclear factories (Figs 6, S7 and S8). Our study reveals that while MT disruption does not prevent core-only factories from transitioning into core+OC intermediate factories (Figs 6, S7, and S8), nocodazole treatment delayed core-to-full virion assembly and led to an ~10-fold reduction in burst size (Fig 8). This delay phenocopied that which occurs when OC µ1 protein levels were reduced using siRNA silencing; in both cases, cores persisted as cores for longer periods before fully assembly (Fig 8E, F, M2-KD and Noco). While nocodazole treatment did not alter the number of LDs or their association with OC proteins (Fig 6E, F), it did result in LDs remaining further from the nucleus and from each other (Fig 6G, H). Given that microtubules are directly involved in the organization and movement of lipid droplets within cells [77], it is plausible that MT disruption decreased full assembly by limiting the availability of OC proteins, similar to the effects of OC protein silencing. Alternatively, microtubules could actively transport cores away from the dense intermediate factories towards the OC proteins, facilitating interactions between cores and OC proteins for full assembly. Additionally, by "parking" fully assembled viruses in perinuclear arrays and removing them from intermediate factories, less crowded spaces may be created to facilitate further OC assembly.

Finally, except for the initial core-only factories comprised primarily of their own cores and proteins (Figs 2 and S3), both intermediate factories and second-generation core-only factories contain a blend of *de novo* proteins from two distinct incoming cores (Figs 4, S5 and S6). In intermediate factories, the presence of mixed-origin proteins is likely a result from the merging of different core-only factories. In contrast, the mixing in *de novo* core-only factories implies an indiscriminate selection of core proteins for creating new factories. While this mixing may be incidental, it could also provide a selective advantage by enabling the co-localization of proteins that compensate for deficiencies caused by mutations. For example, if two co-infecting cores are partially impaired due to mutations affecting core assembly or transcription, mixing could facilitate the production of viable *de novo* cores, increasing the likelihood of generating cores possessing optimal sequences through reassortment. To test this possibility, a strategy would be required to prevent mixing in second-generation core-only factories compared to intermediate factories. Moreover, while it has long been recognized

that co-infecting incoming virions produce genome reassortments, it remains unknown if reassortment occurs in secondary peripheral factories and/or intermediate factories [78–81]. As we gain more understanding of the processes governing the formation, movement, and transition of these factories, it may become possible to develop targeted strategies to understand the contribution of these steps in virus evolution and reassortment.

## Materials & methods

### Cell lines and viruses

BHK-T7 (a generous gift from Ursula Buchholz, NIH), H1299 and L929 cells (purchased from the American Type Culture Collection (ATCC)) were maintained at 37˚C with 5% CO2. BHK-T7 cells were cultured in high-glucose Dulbecco's modified Eagle medium (DMEM) (D5796, Millipore Sigma) supplemented with 10% fetal bovine serum (FBS) (F1051, Millipore Sigma), 2 mM l-glutamine (G7513, Millipore Sigma), 100 U/ml penicillin, 100 µg/ml streptomycin and 25 ng/ml amphotericin B (A5955, Millipore Sigma). Adherent L929s were cultured in Minimal essential medium (MEM) (M4655, Millipore Sigma) supplemented with 5% FBS, 1 × nonessential amino acids (M7145, Millipore Sigma) and 1mM sodium pyruvate (S8636, Millipore Sigma). H1299 cells were cultured in Roswell Park Memorial Institute medium (RPMI) (R8758, Millipore Sigma) supplemented with 10% FBS. L929s cultured in suspension were grown in Joklik's modified MEM (JMEM) (pH 7.4) (M0518, Millipore Sigma) supplemented with 2g/L sodium bicarbonate (BP328, Fisher Scientific), 1.2g/L 4-(2-hydroxyethyl)-1-piperazineethanesulfonic acid (HEPES) (BP310, Fisher Scientific), 1 × non-essential amino acids and 1mM sodium pyruvate. Reovirus serotype type 3 Dearing-PL (T3D$^{PL}$; Dr. Patrick Lee, Dalhousie University), and λ2-FLAG- and λ2-V5-tagged viruses were propagated in suspension L929 cells from a seed stock to preserve genetic identity, extracted with Vertrel XF (Dymar Chemicals) three times, and purified by ultracentrifugation on cesium chloride gradients, as previously described [82].

### Plaque assays

For reovirus titers, plaque assays were done in L929 cells. Reovirus dilutions in serum-free media were added to confluent monolayers of L929 cells for 1hr with gentle rocking every 10 minutes. After 1 hour, an agar overlay was added (1:2 ratio of 2X JMEM media, 1:4 ratio of complete MEM media, and 1:4 ratio of 2% agar). Overlays were allowed to solidify at room temperature for 30 minutes before incubation at 37˚C/5% CO2 for 4–5 days. Then, 4% paraformaldehyde (PFA; in PBS) was added to the overlay for 30mins at room temperature. PFA was then discarded, and agar overlays were gently removed. Cells were then fixed with methanol for 15 minutes at 4˚C. After discarding the methanol, plaques were visualized by staining with a 1% (wt/vol) crystal violet solution (2.5g crystal violet, 50mL methanol, 197.5mL Milli-Q water) and plaques were manually counted.

### Core particle generation, AlexaFluor labelling, and transfections

**Core generation.** Reovirus cores were generated by incubating purified virions with 10µg/mL chymotrypsin (C3142, Millipore Sigma) for 2hours at 37°C in virus dilution buffer (VDB) (150 mM NaCl, 10 mM MgCl2, and 10 mM Tris (pH 7.4)). Following digestion, particles were pellet by ultracentrifugation (100,000xg, 4°C, 90mins). The pellet was then resuspended in 1/10$^{th}$ the original volume in VDB. Confirmation of core generation was performed by SDS-PAGE and Coomassie blue staining (see SDS-PAGE and Western blot section). Particle concentrations were estimated based on Coomassie blue staining and A$_{260}$ readings, where 1 A$_{260}$ is equivalent to 4.4x10$^{12}$ core particles [83].

**Core labelling.** Following core generation, ~1x10$^{12}$ core particles were incubated with 0.05M sodium bicarbonate (pH 8.5) and a 12uM final concentration of Alexa Fluor (AF) 488 (A20000, Thermo Fisher), 546 (A20002, Thermo Fisher) or 647 (A20006, Thermo Fisher) in VDB. The labeling reaction was incubated for 90mins at 4°C on a magnetic rotator and

then dialysis was performed three times (once for 1 hour, once for 3 hours, and once overnight with a molecular weight cutoff of 10–20 kDa) in VDB. Following dialysis, labelled cores were collected and stored at 4°C until use.

**Core transfections.** $4x10^5$ H1299 cells were plated on #1.5 thickness, 18mm glass gelatin-coated coverslips (72222–01, Electron Microscopy Sciences) in 12-well plates and allowed to adhere for ~24 hours. ~$5.5x10^{10}$ labelled core particles were diluted in 37.5μL of OptiMEM (31985–070, Gibco) prior to mixing at a 1:1 volume with diluted Lipofectamine 2000 (11668–019, Invitrogen). Core-Lipo complexes were permitted to form for 20mins before adding dropwise to cells. Cell plates were then centrifuged at 1200xg for 10mins at room temperature, and cells were placed in a 37°C, 5% $CO_2$ incubator until desired timepoints were reached. In experiments where two AF-cores were used, 1 hour after the initial transfection cell media was aspirated, the cells were washed once with PBS, and fresh complete media was added. Then, the transfection protocol outline above was repeated with the second labelled cores, and 1hpt cell media was removed, the cells were washed with PBS, and fresh media was added before placing cells back in the incubator until desired time post-transfection.

## Immunofluorescent confocal microscopy

**Sample preparation.** $4x10^5$ H1299 cells were plated on 18mm glass gelatin-coated coverslips in 12-well plates and allowed to adhere for ~24hrs. For infections, viruses were diluted in serum-free media and added to wells for 1hr at 37°C, with gentle rocking every 10 minutes. Virus media was then removed, replaced with complete media, and cells were incubated at 37°C until the required hpi. Media was removed and the cells were washed with PBS prior to fixation with pre-warmed 4% PFA for 30mins at room temperature. PFA was discarded and the cells were washed 3x for 5 minutes each with PBS. Cells were then permeabilized by incubation with 0.1% Triton X-100 in PBS (PBS-Tx100) for 5mins at room temperature. PBS-Tx100 was then discarded, and cells were washed 3x for 5mins each with PBS. After permeabilization, cells were then incubated with blocking solution (3% NBCS/PBS with 0.1M Glycine) for 1hr at room temperature. Then, blocking solution was removed and cells were incubated with primary antibody solutions (block buffer containing: rabbit polyclonal antibodies against core (α-Core) at 1:1000 (Biologics International Corp), mouse α-σ3 (4F2) at 1:250 (DSHB), rabbit polyclonal antibodies against μNS (α-μNS) at 1:1000 (Biologics International Corp), monoclonal mouse anti-tubulin (12G10, DSHB), monoclonal mouse anti-FLAG M2 (Sigma F1804), and/or polyclonal rabbit anti-V5 (Novus NB600–381)). overnight at 4°C. Primary antibody solution was then removed, and cells were washed with 0.1% Tween-20 in PBS 3x for 5mins each. For secondary antibody staining, antibodies (Alexa Fluor 647 goat α-rabbit, Alexa Fluor 488 goat α-rabbit, Alexa Fluor 647 goat α-mouse, Alexa Fluor 488 goat α-mouse, and Cy3 goat α-mouse) were then diluted (1:300) in blocking solution and cells were allowed to incubate in the solution for 1hr at room temperature while blocked from light. Cellular dyes, such as BODIPY 558 (5μM, D3835, Invitrogen) or BODIPY 493/503 (5μM, D3922, Invitrogen) and DAPI (0.1μg/mL, D1306, Invitrogen were either added in together with the secondary antibody incubation or performed after, following manufacturer instructions. In cases where directly conjugated primary antibodies were used, mouse α-σ3 (10C1 and/or 10G10) directly labelled with AF 647 using the Apex AF 647 Ab-Labelling Kit (A10475, Invitrogen) was added as the last staining step (1:100). Directly conjugated antibodies were made according to the manufacturer's instructions. Coverslips were then mounted onto glass slides using 1 drop (~50μL) of Prolong Diamond AntiFade Mountant (Invitrogen). The edges of the coverslips were then sealed with nail polish. Slides were allowed to dry overnight at room temperature, protected from light, prior to imaging.

*Image acquisition.* All confocal images were captured using the University of Alberta Cell Imaging Core spinning disk confocal microscope (Quorum Technologies) with the following lasers: 405nm, 491nm, 561nm, 640nm and corresponding emission filters for DAPI, GFP, TRITC and Cy5. Images were acquired with a 60X/1.42NA and/or 100x/1.4 oil-immersion lens on a Hammamatsu C9100-13 EMCCD camera (Hamamatsu Corp) using Perkin Elmer's Volocity software.

**Image analysis.** For visual presentation, images were processed from Z-stacked images projected into one 2D-image based on maximum intensities in ImageJ (NIH) software with the Fiji plugin. The brightness of each channel was adjusted

for display purposes only, and images were prepared for figures in Adobe Photoshop. For quantification, 3D-unedited images were used in Volocity software. To quantify the number of a certain object, as well as the objects volume, Volocity compartmentalization was used and to ensure only true factory objects were selected and visual thresholding was performed against background signals. To determine the distance from the nucleus of each object, the DAPI stained nucleus was selected and the edge-to-edge distance from a particular object to the nucleus was then be measured. "Shared" objects were determined to be objects compartmentalized within each other (i.e., when a core object was found within an OC object, FLAG in a V5 object, or vice versa).

## Transmission electron microscopy sample preparation

Cells were cultured on Aclar sheets and chemically fixed using a solution of 2.5% glutaraldehyde and 2% paraformaldehyde in 0.1 M sodium cacodylate buffer containing 2 mM $CaCl_2$ (pH 7.4) for 20 minutes at 37 °C, followed by an additional 40 minutes at room temperature. After removal of the primary fixative and washing with buffer, samples were post-fixed for 1 hour in 1% osmium tetroxide supplemented with 10% potassium hexacyanoferrate(II). Following thorough washing, cells were treated with sodium acetate (pH 5.2) and stained with 2% aqueous uranyl acetate for 1 hour.

Samples were then dehydrated through a graded ethanol series and embedded in Spurr's resin. After complete infiltration, the resin was polymerized at 65 °C for 72 hours. The Aclar sheets were subsequently removed, and 1 × 1 mm resin blocks were excised using a fret saw. These blocks were mounted onto epoxy stubs using cyanoacrylate adhesive and trimmed (Trim90, Diatome). Ultrathin sections (~100 nm) were prepared using a diamond knife (Ultra 45, Diatome).

Sections were post-stained with lead citrate and a second round of uranyl acetate staining, then carbon coated with a Leica EM ACE600 prior to imaging. Micrographs were acquired using a Hitachi H-7650 transmission electron microscope operated at 60 kV, equipped with an AMT digital camera and AMT Image Capture Engine software, as well as on a JEOL JEM-2100 operated at 200 kV, equipped with a Gatan Orius camera and DigitalMicrograph software (Gatan).

## Cloning and recombinant virus generation

**Cloning tagged L2 sequences.** Linear DNA sequences encoding a linker region, restriction enzyme site NcoI, the T3D[PL] L2 gene sequence with a fragment removed and either 2xFLAG or 1xV5 sequences added in place, restriction enzyme site PpuM1, and another linker region were synthesized by Invitrogen (Genestrings). The DNA sequences are listed in Table 2. The fragments were amplified by PCR, products were purified, and then digested with NcoI (FD0575, Thermo) and PpuM1 (FD0764, Thermo) restriction enzymes. The plasmid pBacT7-L2 was similarly digested and gel purified. The fragments were then ligated into the pBacT7-L2 plasmid using T4 ligase (15224–017, Invitrogen). Purified plasmids were sequenced via Sangar Sequencing at The Molecular Biology Service Unit (MBSU) at the University of Alberta.

**Recombinant virus generation.** Recombinant viruses were generated using a plasmid-based reverse genetics system [84]. In short, one 2 cm² well of 100% confluent BHK-T7 cells was transfected with 11 different plasmids encoding the 10 genome segments from T3D[PL] reovirus (pT7-Bac backbone, one with each gene) and C3P3 (pCMVScript backbone, Stratagene) 2.25µg of DNA was added per well, with 6.75µL of TransIT-LT1 transfection reagent (Mirus Bio, MIR 2304) diluted in OptiMEM (31985–070, Gibco). 24hpt, the media was changed. 5 days post transfection virus was collected by scraping cells into PBS, freeze thawed lysates 3x and viruses were titered on L929 cells as previously described.

## SDS-PAGE and Western blot

For standard Western blot analysis, cells were first washed with 1xPBS before lysing with RIPA buffer (50mM Tris pH 7.4, 150mM NaCl, 1% NP-40, 0.5% sodium deoxycholate) supplemented with protease inhibitor cocktail (P8340, Sigma).

**Table 2. FLAG and V5 L2 DNA sequences.**

| DNA Name | Sequence |
|---|---|
| L2FLAG | *GACTAC***CCATGG**AAAAGCGAGGTAACTTCATAGTGGGGCAGAACTGCTCATTAGT AATCCCTGGTTTTAATGCGCAG-GATGTCTTTAACTGTTATTTCAATTCCGCCCTCGCT TTCTCGACTGAAGATGTCAATGCTGCGATGATTCCCCAAGT-GTCTGCGCAGTTTGAT **GGAGACTACAAAGATCACGACGGGGATTATAAGGACCACGATGGA**GAGTGGACG TTGGATATGGTCTTCTCCGACGCAGGAATCTATACCATGCAGGCTCTAGTGGGATCT AATGCTAATCCAGTCTCTTTGGGTTCCTTTGTAGTTGATTCTCCAGATGTAGATATAA CTGACGCTTGGCCAGCTCAGTTAGACTTTACGATCGCGGGAACTGATGTCGATATAA CAGTTAATCCTTATTACCGTCTGATGACCTTTGTAAGGATCGATGGACAGTGGCAGA TTGCCAATCCAGACAAATTTCAATTCTTTTCGTCGGCGTCTGGGACGTTAGTGATGAA CGTCAAATTAGATATCGCAGATA-AATATCTACTATACTATATACGAGATGTCCAGTCT CGAGATGTTGGCTTTTACATTCAGCATCCACTTCAACTTTTGAATAC-GATCACATTGCC AACCAACG*AGGACCTGACTTG* |
| L2V5 | *GACTAC***CCATGG**AAAAGCGAGGTAACTTCATAGTGGGGCAGAACTGCTCATTAGT AATCCCTGGTTTTAAT-GCGCAGGATGTCTTTAACTGTTATTTCAATTCCGCCCTCGCT TTCTCGACTGAAGATGTCAATGCTGCGAT-GATTCCCCAAGTGTCTGCGCAGTTTGAT **GGCAAGCCTATCCCTAACCCTCTGCTGGGGCCTGGACAGCAC-C**GAGTGGACGTTGGAT ATGGTCTTCTCCGACGCAGGAATCTATACCATGCAGGCTCTAGTGGGATCTAATGCTA ATCCAGTCTCTTTGGGTTCCTTTGTAGTTGATTCTCCAGATGTAGATATAACTGACGCT TGGCCAGCTCAGTTAGACTTTACGATCGCGGGAACTGATGTCGATATAACAGTTAATC CTTATTACCGTCTGAT-GACCTTTGTAAGGATCGATGGACAGTGGCAGATTGCCAATCC AGACAAATTTCAATTCTTTTCGTCGGCGTCTGGGAC-GTTAGTGATGAACGTCAAATTA GATATCGCAGATAAATATCTACTATACTATATACGAGATGTCCAGTCTCGAGATGTTG GCTTTTACATTCAGCATCCACTTCAACTTTTGAATACGATCACATTGCCACCAACG*AGG ACCT*GACTTG |

Text Legend: *Linker*, **NcoI**, ***PpuMI***, **FLAG**, V5.

After collection, 5x protein sample buffer (250mM Tris pH 6.8, 5% SDS, 45% glycerol, 9% β-mercaptoethanol, and 0.01% bromophenol blue) was added to a final concentration of 1x. Samples were then boiled at 100˚C for 10 minutes. Samples were electrophoresed on 10% SDS-acrylamide gels, and proteins were transferred from the gel onto nitrocellulose membranes using a Trans-Blot Turbo Transfer System (Bio-Rad). Membranes were incubated with block buffer (3% newborn calf serum (NBCS) in Tris-buffered saline (.05 M Tris and 0.15 M sodium chloride, pH 7.4 with 0.1% TritonX-100 (TBST)) for 1 hour at room temperature, followed by incubation with a primary antibody solution (block buffer containing: rabbit polyclonal antibodies against whole virus (α-Reo) at 1:10,000 (Biologics International Corp), rabbit polyclonal antibodies against core (α-core) at 1:1000 (Biologics International Corp), rabbit polyclonal antibodies against λ2 (α-λ2), monoclonal mouse anti-FLAG M2 (Sigma F1804), and/or polyclonal rabbit anti-V5 (Novus NB600–381)) for either 1 hour at room temperature or overnight at 4˚C followed by a secondary antibody solution (block buffer containing: goat α-rabbit AF 488 or 647 at 1:1000, goat α-rabbit HRP 1:10,000, goat α-mouse AF 488 or 647 at 1:1000, and/or goat α-mouse HRP 1:10,000) for 1 hour at room temperature. Membranes were washed 3x for 5 minutes each in TBST between primary and secondary antibody incubations. Prior to imaging, membranes with HRP-conjugated secondaries were incubated with ECL Plus Western Blotting Substrate (32132, Thermo Fisher Scientific) for 5 minutes at room temperature. Membrane imaging was carried out using a Biorad's Chemidoc Imaging System. Densitometric analysis was performed with ImageJ software, and images were processed for display in Adobe Photoshop. All incubation steps for membranes were done with gentle rocking.

## Binding assays and flow cytometry

To assess reovirus binding, 100% confluent L929s were prechilled for 1hr at 4°C and then infected with serial dilutions of virus at 4°C for 1hr with gentle agitation every 10mins. Three washes with cold PBS were done to remove any unbound virus, and then cells were detached using Cellstripper (25–056-CI Corning). Cells were then fixed with 4% PFA for 30 min at 4°C. Cells were incubated with monoclonal mouse anti-σ3 (1:100, 10C1 or 10G10, DSHB) antibody for 1hr at 4°C, then washed three times with flow buffer (2%FBS, 1mM EDTA in PBS). After each wash, cells were pelleted at 500xg for 5mins

at 4°C. Then, cells were incubated with AF-conjugated goat anti-mouse 647 secondary antibody (1:1000) for 30 mins in the dark at room temperature. Samples were analyzed using BD LSR Fortessa (BD Biosciences), and a minimum of 10,000 cells were collected for each sample. Data was analyzed using FlowJo v10 Software (BD Biosciences)

## Nocodazole and cycloheximide treatments

80-90% confluent H1299 cells were infected as indicated in specific experiments. In short, cell media was removed, cells were washed once with PBS, and virus diluted in serum-free media was added to the cells for 1 hour with gentle rocking every 10 minutes. One hour later, virus media was removed, and complete media containing 10µM nocodazole, or an equivalent volume of DMSO, was added to the cells. Cells were then incubated until desired time points were reached. Where indicated, cycloheximide (100 µg/mL) was added to complete media containing nocodazole or DMSO as well to inhibit translation and thus synthesis of *de-novo* viral particles.

## DsiRNA-mediated knockdowns

For DsiRNA knock-down studies, cells were transfected with DsiRNAs (Integrated DNA Technology) directed against either a non-targeting control (G7L, vaccinia virus gene, Antisense Sequence: rUrUrUrArUrUrUrGrArUrGrArArUrCrUrAr-GrUrUrGrGrUrUrCrUrC Sense Sequence: rGrArArCrCrArArCrUrArGrArUrUrCrArUrCrArArArUrAAA), or M2 (reovirus µ1 protein, Antisense Sequence: rGrArCrCrCrUrGrArGrArUrGrArArUrUrArUrUrArUrCrUTT Sense Sequence: rArArArGrAr-UrArArUrArArUrUrCrArUrCrUrCrArGrGrGrUrCrArG). Cells were plated and transfected simultaneously by collecting cells via trypsinization and diluting to $2x10^5$ cells/mL and mixing with a pre-incubated cocktail of 1:2 volume Lipofectamine 2000 (11668–019, Invitrogen) to 2mM DsiRNA in Opti-MEM (31985–070, Gibco) prior to plating 1mL/well per 4 cm2. Cells were allowed to adhere for at least 6 hours prior to infection.

## RT-qPCR

Cells were lysed in TRI Reagent (T9424, Millipore Sigma) and the aqueous phase was separated following chloroform extraction as per the TRI Reagent protocol. Isopropanol was mixed with the aqueous phase and RNA was isolated as per the Monarch Total RNA Miniprep kit (T2010S, New England Biolabs) protocol. RNA was eluted using RNAse free water and total RNA was quantified using a NanoDrop spectrophotometer. Using 200ng RNA per 12µl reaction, cDNA synthesis was performed with random primers (48190011, ThermoFisher Scientific) and M-MLV reverse transcriptase (28025013,

Table 3. Gene specific primers used in RT-qPCR reactions.

| Primer Name | Sequence |
| --- | --- |
| S2RTfwd1 | ACGCTTAGTGTGGTCAGCTC |
| S2RTrev1 | TGAATCTTGGATCACGCGCT |
| S4negfwd | GGAACATTGTGAGAGCAGCA |
| S4negrev | GCAAGCTAGTGGAGGCAGTC |
| M1RTfwd1 | CAACGTTGATCGTCGGCTTC |
| M1RTrev1 | GAGAGGTGCGTAGACATCCG |
| M2RTfwd1 | AATCAGCCTTGGTGCCCTAC |
| M2RTrev1 | CTGACAGCACACGCATCTTG |
| L1RTfwd1 | GGGATTGCGAAATCAGGTGC |
| L1RTrev1 | GCACTACCAGACGTGGTTGA |
| L3RTfwd1 | CCCCGATGCTGAGAAATGGT |
| L3RTrev1 | TGCTCGATCAAACCGTCCAA |

ThermoFisher Scientific) as per the manufacturers protocol. Following a ¼ cDNA dilution, RT-PCR reactions were executed following the SYBR Select (4472920, Invitrogen) protocol using reovirus gene-specific primers (listed in Table 3) and the CFX96 system (Bio-Rad). All RT-qPCR reaction plates included a no template and no reverse transcription control.

## Supporting information

**S1 Fig. AF-labelled cores are replication competent and demonstrate similar single-step growth kinetics compared to non-labelled cores.** T3D$^{PL}$ virions were digested with chymotrypsin to generate core particles in-vitro. A subset of cores was then labelled with AF-488, -564, or -647 dyes. (A) Whole virus (virus), non-labelled core particles (core) and labelled cores (AF-cores) were subject to SDS-PAGE. The gel was then immediately imaged under AF-488, Cy3, and AF-647 filters using BioRad's ChemiDoc MP imaging system. Then, the same gel was stained with Coomassie and imaged again for total protein identification. (B) The ratio of non-labelled cores to volume of Lipofectamine 2000 transfection reagent was assessed in H1299 cells. 18 hpt, cells were fixed and immunostained using polyclonal sera from rabbits immunized with reovirus particles (α-Reo, red) and stained with DAPI to visualize nuclei. (C) ~1000 non-labelled (Cores, black line, n = 5) AF-Cores (blue line, n = 3) per cell were transfected into H1299 cells. Lysates were collected every 3 hours for 12 hours and virus production was assessed via plaque assay. Data is graphed as the mean +/- SD at each timepoints. Statistical analysis is reported as two-way ANOVA with Sidak's multiple comparisons test between the mean of each group at their respective timepoints. ****$p < 0.0001$, ***$p < 0.001$, **$p < 0.05$, ns > 0.05. *In vitro* core digestions can leave some intact whole virions that produce some background fluorescence when immunostaining for de novo virus proteins in no-lipofectamine conditions. Such whole virions however would not have AF-labelled cores, and the potentially AF-labelled outercapsid proteins would be shed during endocytosis. Nevertheless, we used conditions that provided notable increase in fluorescence in the presence of lipofectamine over the absence of lipofectamine, to skew towards a majority proportion of AF-core-driven infection.
(TIF)

**S2 Fig. Input cores move from OC(-)-peripheral to OC(+)-perinuclear factories and seed de-novo OC(-)-factories in the periphery.** H1299 cells were transfected with ~1000 AF 647-labelled reovirus core particles per cell (magenta). At (A) 6, (B) 9 and (C) 12 hpt cells were fixed and subsequently immunostained with rabbit polyclonal sera generated against μNS (AF 488, green), mouse monoclonal anti-σ3 (10G10, Cy3, yellow) and nuclei were stained with DAPI (blue). All images captured by spinning disk confocal microscopy and are images of compressed Z-stacks. White asterisks indicate aggregates formed by AF-cores.
(TIF)

**S3 Fig. Input cores establish independent peripheral compartments.** H1299 cells were first transfected with ~1000 AF-564 cores per cell. One hour later, the cells were transfected with ~1000 particles per cell of AF-647 labelled cores. 7 hpt, cells were fixed and processed for immunofluorescence confocal microscopy imaging. Representative images of compressed Z-stacks. White asterisks indicate aggregates formed by AF-cores. **(A)** Cells were stained with DAPI to visualize nuclei. **(B)** Cells were not stained with DAPI.
(TIF)

**S4 Fig. Generation of recombinant L2-tagged viruses.** Schematics demonstrating amino acid regions tested for incorporation of a 2xFLAG tag or 1xV5 tag. Colours correlate with those used in Table 1. (A-E) Modeling of λ2 attempted tag regions (structures from PDB accession #1EJ6) in (A) the λ2 monomer, (B) the λ2 pentamer, (C) the asymmetric core unit, (D) the whole assembled core, and (E) a zoomed in pentamer assembled within the core. (F-H) Modeling of the λ2 attempted tag regions (PDB #2CSE) in (F) the whole virion asymmetric unit, (G) the whole assembled virion, and (H) a zoomed in pentamer assembled within the whole virus. (I) Cartoon schematic showing the tagged viruses generated in

the reverse genetics system, with tagged proteins colour coded. These are the same icons shown in Figure 3B, created in BioRender. *Shmulevitz*, M. (2025) https://BioRender.com/3n918ke.
(TIF)

**S5 Fig. Majority of independent input core containing OC(-) factories do not obtain de-novo core proteins from non-self factories.** (A-B) H1299 cells were first transfected with ~1000 AF-546 labelled FLAG- or V5-tagged cores per cell. One hour later, the cells were transfected with ~1000 particles per cell of AF-647 labelled FLAG- or V5-tagged cores (opposite tag to the first transfection). 7 hpt, cells were fixed and processed for IF-CM. Representative images of compressed Z-stacks. White asterisks indicate aggregates formed by AF-cores. (A) Cells were immunostained with monoclonal mouse α-FLAG (AF 488, green) and stained with DAPI to visualize nuclei. (B-C) Cells were immunostained with α-FLAG (AF 488, green). (C) H1299 cells were first transfected with ~1000 particles per cell of AF-647 labelled FLAG- or V5-tagged cores. 1 hour later, the cells were transfected with ~1000 particles per cell of AF-546 labelled FLAG- or V5-tagged cores. 7 hpt, cells were fixed and processed for IF-CM. Representative images of compressed Z-stacks. (D and E) Graphs represent the same data displayed in Figure 4C but split to represent (D) data sets from where AF-FLAG particles were transfected first, and (E) where AF-V5 particles were transfected first. Similarly, (F and G) graphs represent the same data displayed in Figure 4D but split to represent (F)) data sets from where AF-FLAG particles were transfected first, and (G) where AF-V5 particles were transfected first. Data is plotted as mean +/- 95% CI. Statistical analysis is reported as ordinary one-way ANOVA between the mean of each column. ****$p < 0.0001$, ***$p < 0.001$, **$p < 0.05$, ns $> 0.05$.
(TIF)

**S6 Fig. *De-novo* core proteins start to mix indiscriminately between independent compartments created by input cores.** Immunofluorescent images captured via spinning disk confocal microscopy. H1299 cells were co-infected with λ2-FLAG and λ2-V5 at an MOI of 5 each. 8 hpi, cells were fixed and immunostained with (A) monoclonal mouse α-FLAG (AF 488, green), polyclonal rabbit α-V5 (AF 647, magenta) and monoclonal mouse α-σ3 directly conjugated to Alexa Fluor 594 (10C1, yellow) and stained with DAPI for nuclei visualization (blue). (B) The secondary antibodies were swapped from (A), and α-FLAG staining was coupled with AF 647 (magenta) and α-V5 was coupled with AF 488 (green).
(TIF)

**S7 Fig. More image examples of microtubule depolymerization diminishes perinuclear factories and stalls OC(+) factories in intermediate regions.** H1299 cells were infected with T3D^PL MOI 3. 1 hpi, complete media containing 10μM nocodazole, or an equivalent volume of DMSO (control), was added to the cells. 8 or 12 hpi, cells were fixed and immunostained for immunofluorescence confocal microscopy imaging. Compressed Z-stack images of cells stained with: (A) monoclonal mouse anti-Tubulin (12G10, AF 647, magenta), monoclonal mouse anti-σ3 (10C1, directly conjugated to AF 594, yellow), polyclonal rabbit antibodies raised against reovirus cores (AF 488, green), and DAPI for nuclei visualization (blue). (B) Tubulin (12G10, AF 647, magenta), BODIPY 493/503 for lipid droplet visualization (LDs, yellow), and DAPI for nuclei visualization (blue). (C) Monoclonal mouse anti-σ3 (10G10 followed by AF 647 conjugated secondary antibodies (magenta)), BODIPY 493/503 for lipid droplet visualization (LDs, yellow), and polyclonal rabbit antibodies raised against reovirus cores (AF 488, green).
(TIF)

**S8 Fig. The localization of input AF-core particles is not microtubule dependent.** H1299 were transfected with ~1000 AF-546 reovirus core particles per cell (magenta). 1 hpt, cells were treated with (A) DMSO or (B) 10μM nocodazole. 12 hpt, cells were fixed and immunostained with monoclonal mouse antibodies directed against σ3 (10G10, α-σ3, AF 647, cyan) polyclonal rabbit antibodies raised against reovirus cores (α-core, AF 488, green) and DAPI was used to stain nuclei. White asterisks indicate aggregates formed by AF-cores. All images are of compressed Z-stacks captured by IF-CM.
(TIF)

**S9 Fig. The role of microtubules in intermediate to perinuclear factory transition is not limited to lab-adapted reo-viruses.** (A) H1299 cells were infected with reovirus ISVP strains Type 3 (T3DS1-T249I), Type 2 (T2E1, T2E2), or Type 1 (T1L, T1E1) to give 80% infection of cells by 12 hpi. 1 hpi, media containing either 10μM nocodazole or an equivalent volume of DMSO were added to the cells. 12 hpi, cells were fixed and immunostained with monoclonal anti-σ3 (10G10, AF 647, magenta) and polyclonal rabbit antibodies raised against reovirus cores (AF 488, green), and stained with BODIPY 493/503 for lipid droplet visualization (LDs, yellow) and DAPI for nuclei visualization (blue). Images are representative immunofluorescence confocal microscopy Z-stacks, of n = 5–10 images per virus type. (B) M1 gene sequences were acquired from NCBI Genbank and aligned using ClustalOmega with default settings. Alignments were then viewed using SnapGene software. Amino acids at position 208 are highlighted in blue.
(TIF)

**S10 Fig. Microtubules contribute to virion organization in intermediate and perinuclear factories.** H1299 cells were infected with T3D$^{PL}$ MOI 3. 1 hpi, complete media containing 10μM nocodazole, or an equivalent volume of DMSO was added. Cells were fixed and processed for transmission electron microscopy analysis as described in the methods section at 18 hpi. Select factory regions showcasing the arrangement of cores, full virions, empty virions, and OC shells are shown in the boxed regions.
(TIF)

## Acknowledgments

We thank all members of the Shmulevitz lab for their invaluable feedback and suggestions for this project. We also thank Dr. Hilmar Strickfaden and Kiera Howie for access to the state-of-the-art microscopy facility, for their generous use of time as technical support for all microscopy-based experiments.

## Author contributions

**Conceptualization:** Justine Kniert, Maya Shmulevitz.

**Data curation:** Justine Kniert, Dante Terino, Heather E. Eaton, Maya Shmulevitz.

**Formal analysis:** Justine Kniert, Dante Terino, Heather E. Eaton, Maya Shmulevitz.

**Funding acquisition:** Maya Shmulevitz.

**Investigation:** Justine Kniert, Dante Terino, Heather E. Eaton, Shiau-Yin Wu, Hilmar Strickfaden, Maya Shmulevitz.

**Methodology:** Justine Kniert, Heather E. Eaton, Qi Feng Lin, Shiau-Yin Wu, Hilmar Strickfaden, Maya Shmulevitz.

**Project administration:** Justine Kniert, Maya Shmulevitz.

**Resources:** Heather E. Eaton, Shiau-Yin Wu, Hilmar Strickfaden, Maya Shmulevitz.

**Supervision:** Maya Shmulevitz.

**Validation:** Justine Kniert, Dante Terino, Heather E. Eaton, Maya Shmulevitz.

**Visualization:** Justine Kniert, Dante Terino, Maya Shmulevitz.

**Writing – original draft:** Justine Kniert, Maya Shmulevitz.

**Writing – review & editing:** Justine Kniert, Dante Terino, Heather E. Eaton, Qi Feng Lin, Maya Shmulevitz.

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
