## [Decision Letter · Decision Letter 0]

1 Jul 2025

PPATHOGENS-D-25-01291

Spatiotemporal coordination of reovirus peripheral core replication to perinuclear whole virus assembly

PLOS Pathogens

Dear Dr. Shmulevitz,

Thank you for submitting your manuscript to PLOS Pathogens. After careful consideration, we feel that it has merit but does not fully meet PLOS Pathogens's publication criteria as it currently stands. Therefore, we invite you to submit a revised version of the manuscript that addresses the points raised during the review process.

Please submit your revised manuscript within 30 days Aug 30 2025 11:59PM. If you will need more time than this to complete your revisions, please reply to this message or contact the journal office at plospathogens@plos.org. Please include the following items when submitting your revised manuscript:

We look forward to receiving your revised manuscript.

Kind regards,

John S.L. Parker, DVM, Ph.D.

Guest Editor

PLOS Pathogens

Sonja Best

Section Editor

PLOS Pathogens

Sumita Bhaduri-McIntosh

Editor-in-Chief

PLOS Pathogens

orcid.org/0000-0003-2946-9497

Michael Malim

Editor-in-Chief

PLOS Pathogens

orcid.org/0000-0002-7699-2064

**Additional Editor Comments:**

The four reviewers all agreed that this is an interesting paper that uses innovative experimental approaches to track the spatiotemporal formation of reovirus factories and the distribution of outer capsid proteins. The paper is well written, and the data are clear. The strengths identified by the reviewers are: (i) the careful and well-controlled experiments, (ii) the compartmentalization of outer capsid negative incoming core-derived VFs from intermediate Core + OC factories, and perinuclear VFs containing assembled virions (iii) the interesting findings regarding the dynamics of viral factory formation and the role of microtubules in regulating the recruitment of outer capsid proteins to viral factories later in infection, and (iv) the novel approach used to track the derivation of de novo synthesized core proteins using the tagged lambda2 proteins. In sum, all reviewers were impressed with the work and the findings. However, three of the four reviewers noted some weaknesses: (i) some of the conclusions are extensions or were derived from the authors previous publication and so are not in themselves new, (ii) the mechanisms underlying the findings described by the authors have not as yet been identified, (iii) one reviewer was concerned about the aggregation of transfected core particles - this seems possible given that core particles are relatively hydrophobic compared to ISVPs and virions, and (iv) some concern about the use of the appropriate statistical analysis.

I also read the paper and found the data and the model presented to be compelling. I think what is missing is an experiment that tests one of the model's predictions. My reading of the reviewers' comments is that this is what would elevate the paper. Although the nocodazole treatment tests the role of MTs, similar experiments have been reported by others, and the current findings clarify rather than reveal. Although I think the paper would be considerably strengthened by testing one of the predictions made from the model they have presented, in the end, I believe that the quantity and quality of the data presented are sufficient. The descriptive model significantly advances our understanding of reovirus replication. I would therefore ask the authors to address all reviewers' critiques in writing and to incorporate changes to the manuscript that acknowledge the gaps and caveats in their findings.

**Journal Requirements:**

At this stage, the following Authors/Authors require contributions: Justine Kniert, Dante Terino, Heather E. Eaton, Qi Feng Lin, and Maya Shmulevitz. Please ensure that the full contributions of each author are acknowledged in the "Add/Edit/Remove Authors" section of our submission form.

2) Some material included in your submission may be copyrighted. According to PLOSu2019s copyright policy, authors who use figures or other material (e.g., graphics, clipart, maps) from another author or copyright holder must demonstrate or obtain permission to publish this material under the Creative Commons Attribution 4.0 International (CC BY 4.0) License used by PLOS journals. Please closely review the details of PLOSu2019s copyright requirements here: PLOS Licenses and Copyright. If you need to request permissions from a copyright holder, you may use PLOS's Copyright Content Permission form.

Potential Copyright Issues:

i) We note that Figures 1. 2, 3, 4, 5, 8, and S4. are created through BioRender. Please confirm that you hold a Premium account and provide a pdf copy of the CC BY 4.0 Licence as provided by BioRender. For instructions on how to generate a CC BY 4.0 license for your figure, please see the guidelines here: https://help.biorender.com/hc/en-gb/articles/21282341238045-Publishing-in-open-access-resources. 

If you are using the free assets from BioRender, we are unable to publish these images as they are licenced under a stricter licence than CC BY 4.0. In this case we ask you to remove the BioRender images and replace them with open source alternatives.

See these open source resources you may use to replace images / clip-art:

- https://bioart.niaid.nih.gov/ 

- https://bioicons.com/

- https://healthicons.org/ 

- https://scidraw.io/

- https://reactome.org/icon-lib

- https://www.phylopic.org/images 

- https://journals.plos.org/plosbiology/article?id=10.1371/journal.pbio.3002395

3) We note that your Data Availability Statement is currently as follows: "All relevant data are within the manuscript and its Supporting Information files.". Please confirm at this time whether or not your submission contains all raw data required to replicate the results of your study. Authors must share the “minimal data set” for their submission. PLOS defines the minimal data set to consist of the data required to replicate all study findings reported in the article, as well as related metadata and methods (https://journals.plos.org/plosone/s/data-availability#loc-minimal-data-set-definition).

4) Please ensure that the funders and grant numbers match between the Financial Disclosure field and the Funding Information tab in your submission form. Note that the funders must be provided in the same order in both places as well. Currently, the Financial Disclosure states there was no funding received.

**Reviewers' Comments:**

Reviewer's Responses to Questions

**Part I - Summary**

Reviewer #1: The authors investigate the dynamics of replication complexes during mammalian reovirus infection. Using innovative approaches to track the components of replication complexes, including fluorescently labeling reovirus core particles and transfecting them into cells to initiate infection and engineering viruses with different epitope tags in the lambda2 protein to observe localization of de novo produced proteins, coupled with advanced microscopy, the authors demonstrate that the initial cores deposited into the cell form distinct “factories” that migrate from the cell periphery toward the nucleus over time. These factories are initially devoid of outer capsid proteins, but over time as they migrate to the cell interior, they pick up the outer capsid proteins and assemble into full particles. This initial migration is not dependent on microtubules, in contrast to the later stages of deposition of particles into the perinuclear region, which does require microtubule scaffolding. Interestingly, the initial factories are more likely to contain de novo-synthesized proteins derived from the “self” cores, whereas later in the process, the factories are more likely to include proteins encoded from other cores.

Overall, the manuscript is clear and the results are innovative and novel. The experiments are well-controlled and the data generally supports the conclusions drawn by the authors. There are no major concerns with the manuscript – very nice job! Some minor concerns are outlined below.

Reviewer #2: The current study is an extension of a previous study that described temporospatial compartmentalization to orchestrate reovirus core versus outer capsid assembly. In the current study, large amounts of data were collected from immunofluorescence confocal microscopy images, and clever approaches were used to differentially label and simultaneously image nearly identical virus particles or proteins. The study validates several previous ideas and provides some new insights into the dynamics of reovirus replication.

In their previous publication, the authors showed that core amplification occurs in peripheral outer capsid (OC)(-) factories early in infection, and OC proteins associate with lipid droplets. They showed that OC(+) factories move towards the nucleus as new OC(-) factories form in the periphery and that late in infection, whole virus particle assembly occurs in perinuclear factories. In the current study, the authors used a combination of human cell infection and transfections with reovirus cores, in some cases labeled with distinct fluorophores. At early or late time points, they immunostained to detect core proteins or OC proteins, nuclei, lipid droplets, or microtubules, and analyzed the cells using immunofluorescence confocal microscopy. They engineered reoviruses containing a V5 or FLAG epitope on a loop in a core protein to enable simultaneous core protein detection. They found that early in infection, most OC(-) factories were peripheral and core(+), but over time, more core(+) factories were OC(+) and located closer to the nucleus, while new OC(-) factories were detected in the cell periphery. These results are consistent with the previous findings and indicate that new factories are seeded into the cell periphery as core(+) factories move towards the nucleus. Following transfection with differentially labeled cores, most factories were positive for only one fluorophore at early times. Although input cores often led to the formation of de novo factories that contained no cores, core(+) factories more often contained proteins from a self core than a non-self core. These observations suggest that peripheral factories often form around input cores. Following infection, both peripheral OC(-) factories and OC(+) factories closer to the nucleus that were both V5(+) and FLAG(+) were larger than those positive for only a single fluorophore. These findings suggest that larger, more mature factories form from the merger of independent peripheral factories, which is consistent with reported observations for reovirus factories and the liquid condensates that form them.

Other groups previously showed that cell treatment with nocodazole impaired reovirus perinuclear factory accumulation, and the authors showed that knockdown of microtubule-interacting OC protein mu1 prevented accumulation of perinuclear factories. Consistent with these observations, in the current study, treatment with nocodazole, which depolymerizes microtubules, had no effect on factory formation or core association with factories, but it impaired OC(+) perinuclear factory accumulation and delayed infectious virus production. The authors found that there was a delay in virus titer relative to viral (+)RNA accumulation in M2 siRNA– and nocodazole-treated cells compared with DMSO-treated cells, which they interpreted as a delay in whole virus production relative to core amplification.

The current manuscript is well written, and many of the authors’ conclusions are well supported. The inclusion of many images, rather than just one or two representatives, and numerous quantified factories and cells instills confidence in the results. Key insights from the current study are largely descriptive rather than mechanistic, but they do enhance an understanding of a dynamic and complex process. Although many smaller nuanced conclusions can be drawn, big takeaways are that many peripheral factories initially form around input cores and contain proteins matched to the input core, but new core(-) peripheral factories often contain mixtures of proteins from different input cores. Although the migration of OC(+) factories into immediately perinuclear depots depends on microtubules, migration of peripheral OC(-) into intermediate areas and formation of OC(+) factories does not. Weaknesses include the more descriptive and validating nature of some findings and a few issues with experiments approach or interpretation. One issue is the delivery of labeled cores via lipofection, which is necessary to initiate infection but often leads to core aggregation and might result in different outcomes than the natural route of infection. Others are the way differences in perinuclear factories and cores versus whole virus are quantified, which could be improved to strengthen conclusions about the contributions of microtubules to virus factory and particle maturation.

Reviewer #3: In this paper, Kniert et al. attempt to elucidate the dynamics of reovirus core replication and viral particle assembly. Using transfection of in vitro-generated cores, the authors demonstrate that cores initially form factories in the periphery of cells. These factories lack outer capsid proteins. During the course of infection, these cores also seed new factories in the periphery while they themselves move toward the nucleus. Different cores introduced into the same cell form independent factories that can eventually merge with each other. These processes all occur independently of an intact microtubule network. Yet, microtubules are required for the formation of new virions from cores.

The experiments performed by the authors are cleverly designed, well-controlled, and effectively executed. They support the conclusions made by the authors. In the opinion of this reviewer, the studies are: (1) a descriptive account of what happens in the cells when cores enter but lack any experimentation to conclude how such things occur (Major Comment A); (2) make conclusions whose relevance to infection isn’t obviously clear (Major Comment B); and (3) either show negative data or only corroborate previous conclusions without reconciling possible differences with previous manuscripts (Major Comment C).

Major Comment A: It is clear that input cores form factories in the periphery that eventually transition toward the nucleus and acquire outer capsid proteins. This result is reminiscent of the paper by the same authors that demonstrated this for steps that occur later in infection. We didn’t know then, and we don’t know now, whether the peripheral location preference of cores is an active process, how it’s mediated, or what mediates their transport to the center of the cell toward the nucleus. It is also not clear from the experiments presented here whether translation occurs in the vicinity of the input core, or whether viral proteins that are synthesized elsewhere are being recruited to the input core and only visible there due to increased local concentration.

Major Comment B: Input cores form one-hundredth or one-thousandth the number of cores that will exist in cells in a few hours. It’s not clear why it’s important to know where input cores go after infection is established. What is the consequence of not establishing a peripheral factory, of two cores not forming independent factories at the beginning, or of temporally changing when they move toward the nucleus and merge? Without that information, all we know is that all this happens in this situation—not why it might be important that this happens. Its is known from this authors' and others' work that virus strains differ in their properties. And if they differ, does it relate to any phenotypes? I do want to acknowledge here that the generation of tagged lambda2 virions is a herculean task, but I am not sure their use here is providing any information that is field-altering. Similarly, the use of two different AF-cores is clever.

Major Comment C: To address my comment in A, the authors do attempt to look at the role of microtubules in this process. While movement of early cores is unaffected, the later steps in infection—the transition of cores to whole virions—are affected, and this phenotype resembles what might happen if the outer capsid proteins were not expressed. We don’t necessarily know from the experiments presented why this is the case. Further, the previous publication (#58 in manuscript) indicated that microtubules are needed for efficient genome packaging. It is not clear from the data presented here if the reduction in titer occurs due to absence of genome packaging or if assembly of the outer capsid on the particle is also compromised when the microtubules are disrupted. Since burst size is measured by PFU, it’s hard to know if one or two defects occur when microtubules are disrupted.

Reviewer #4: This manuscript by Kniert et al. presents novel and conceptually interesting findings on reovirus replication dynamics. The authors employ elegant virological and cell biological approaches to demonstrate that initial input cores establish distinct replication factories. Central to their strategy is the engineering of recombinant reoviruses carrying different antigenic tags (V5 and FLAG) on the core protein λ2, allowing the visualisation of transcriptionally active input cores in situ. These modified viral cores represent a valuable new tool for studying spatiotemporal aspects of reovirus replication and organisation, and they open exciting possibilities for future investigation into the formation and identity of viral replication compartments. This is an exciting study that advances our understanding of reovirus replication compartmentalization. With the addition of experiments to directly test the dynamic behaviour of input cores and minor revisions to improve clarity, the manuscript would be strengthened considerably. I look forward to seeing this work published.

Major Comments:

A key finding is that AF-labelled input cores accumulate over time within OC-containing replication factories. This raises the interesting hypothesis that input cores transition into or merge with these compartments during infection. To support this dynamic model more robustly, I strongly encourage the authors to consider live-cell imaging approaches to directly track these transitions. If such experiments are not feasible due to technical limitations, alternative strategies could be explored. For example, treating cells with aliphatic diols such as 4% 1,6-hexanediol or 5% propylene glycol, which are known to transiently dissolve membraneless organelles, might allow the reversible dispersal of these replication factories. Tracking the redistribution of AF-labelled particles before and after such treatment could provide important insights into the plasticity and dynamics of these viral condensates.

Additionally, the authors could consider using AF-labelled cores (different colors), and upon brief treatments with aliphatic diols followed by fixation, co-localization of cores in re-formed factories could be quantified. This would help determine whether the initial segregation of input cores is actively maintained or whether they can mix upon condensate dissolution. Such experiments would also offer an opportunity to estimate what proportion of cores originally co-localize versus re-associate stochastically upon factory reassembly.

In general, the data are compelling, though a more direct demonstration of factory plasticity and input core behaviour over time would significantly enhance the manuscript.

**Part II – Major Issues: Key Experiments Required for Acceptance**

Reviewer #1: N/A

Reviewer #2: Labeling the cores with distinct fluorophores, as in Fig. 2, is a clever strategy. However, the cores are being delivered by lipofection, and in many cases they form large aggregates. Is there any evidence that cores aggregate following high multiplicity infection? The authors should provide some evidence that this delivery method is relevant when trying to make conclusions about whether cores primarily establish independent puncta during natural infection.

Although factories do appear farther from the nucleus and more dispersed in nocodazole-treated cells in Figs. 6 and S7, perinuclear ‘ringing’ does not appear to be a consistent or definitive OC(+) factory trait at 12 h p.i. in DMSO-treated cells. Is there a different way to quantify the suggested differences between perinuclear OC(+) factories in DMSO- and nocodazole-treated cells, such as interconnectedness or sphericity of the factories?

The authors conclude that nocodazole treatment interferes with the transition of reovirus particles from cores to whole virions. However, the concentration of +RNA is used as a proxy for the number of cores per cell, and the titer of infectious virus is used as a proxy for the number of whole particles. Although this approach is reasonable in a certain light, it is unclear whether all cores transcribe with the same efficiency, and it is known that reovirus can have a high particle-to-PFU ratio. So, these are not optimal proxy measurements. The conclusion should be strengthened by providing evidence that these are relevant proxies, quantifying in a different manner, or using a more direct approach, such as electron microscopic imaging.

Reviewer #3: I feel either A, B or C need to be better addressed through experimentation.

Reviewer #4: No major requests, I have suggested some additional experiments that could enhance the manuscript (please see above).

**Part III – Minor Issues: Editorial and Data Presentation Modifications**

Reviewer #1: 1) Line 75 – Use the abbreviation “LD” here, since it is the first time lipid droplets are discussed and the LD abbreviation is used subsequently throughout.

2) Fig. 1. Multiple unpaired t-tests are not appropriate for multiple variable analysis, especially for percentage-based data. Please revise accordingly.

3) Fig. 1G is not referenced in the results text

4) Fig. 3D: It would be helpful to show a representative blot, from which the quantification was drawn. Additionally, demonstrating that co-infection also results in similar protein expression (and titers) would help solidify that there is not any unanticipated effect of coinfection (superinfection exclusion?) on the results.

5) Line 274: a “majority” would imply greater than 50%, not 47% - a “plurality” may be a more precise term?

6) Fig. 4E: the label for “de novo l2-FLAG proteins” is cut off

7) Fig. 8B: The bands (and which antibodies were used) are not labeled on the gel. The “S” in “Sample” is cut off

8) Fig. 8E-F: the rationale for cycloheximide treatment/normalization should be explained in the results text, and included in the methods section.

Reviewer #2: Fig. S1B. Since there is some background in ‘no lipofectamine” fields, there are likely some contaminating virions in the core preparations.

Fig. S1C. What might account for the difference in titer at 3h p.i.? A difference in binding? Also, was the correct statistical test applied? Likely, ANOVA should be used with post-hoc tests if the difference is significant.

Fig. 1D-E. There are no statistical comparisons shown for the same timepoints between OC+ and OC- factories.

Fig. 1G. Is a similar analysis needed for % of OC(+) factory objects with or without AF-Cores out of total OC(+) objects in order to make the point made in lines 363-366? A late timepoint is not examined elsewhere before this statement.

Fig. 4C-D. Although there does not appear to be a visual difference, the data points for ‘added first’ and ‘added second’ should be split out into separate graphs and statistically analyzed separately to determine whether order of addition influences association of the tagged core with the tagged protein.

Fig. 5A. There appears to be very little colocalization of sigma3 with V5 or FLAG, and sigma3(+) puncta are quire small. This image does not appear representative of most of the images in Fig. S6 or the graphed data. Is there a reason why it was selected?

Line 424-426. The factories certainly appear more dispersed after nocodazole treatment. However, there is no reported difference in size for OC(+) factories. And the authors haven't shown in this study that the depots are full of assembled particles (or that those in noco-treated cells are not). So, this conclusion is a bit of a stretch and should be rephrased.

Line 429-430, the authors suggest that OC(+) factory progression is stalled. In what way? At this point in the manuscript, what evidence suggests being close to the nucleus is required for replication?

Fig. S9. Since all factories examined were for reoviruses containing P208, it might be important to determine whether the phenotype holds for factories containing S208 in mu1. In other words, is mu1 association with microtubules important for efficient whole virion assembly?

Fig. 8E-F. If plotted on the same graph, would there appear to be a decrease in core amplification following nocodazole treatment for Fig. 8E left and right panels? Is the ratio of core:virus simply the ratio of viral +RNA concentration:plaque titer? If so, is this a legitimate method of quantifying particles?

Reviewer #3: NA

Reviewer #4: Minor Comments:

• Page 2: The term "de novo" cores could be more clearly referred to as "nascent" cores, which is more standard in the literature.

• Line 55: Please clarify "each of 12 vertices" by specifying that this refers to the twelve vertices of the icosahedral viral particle.

• Line 60: The term "pseudo-organelles known as condensates" would be clearer and more consistent with the broader field if replaced with "membraneless organelles formed through phase separation (also referred to as condensates)."

• Line 113: The description of OC proteins being "stored at lipid droplets" may imply an active storage mechanism. A more accurate phrasing might be that these proteins "co-localise with lipid droplets."

• Line 143: In the main text, "OC-negative factories" are mentioned, while in Figure 1 the factories are identified using σ3 staining. Please ensure consistent nomenclature between text and figure legends.

• Line 179: The statement about "a single conjugated fluorophore" per particle needs clarification. Was this directly measured or inferred based on labelling stoichiometry? If the estimation is based on dye-to-particle molar ratios, please discuss potential issues such as incomplete labelling and the presence of unlabelled particles in the preparation.

• Line 205: The term "non-self factories" in the subheading is somewhat ambiguous. Consider rephrasing for clarity, perhaps specifying the nature of these compartments more explicitly.

• Table 1: As a stylistic suggestion, the coloured table may be better placed adjacent to the relevant figure. Removing colours (particularly for viruses that could not be rescued) might improve readability. Unrescuable constructs could instead be noted in the figure legend or table text.

• Line 255: Please revise "insufficient" to "insufficiently exposed"

• Line 264: The phrase "the majority (~56%)" could be more precisely phrased as "just over 50%" to avoid the implication of a more substantial majority.

• Line 358: Figure 9 is referenced before Figures 6, 7, and 8. Please check and revise figure callouts to reflect their proper order in the text.

• Figure 7: Please indicate the number of cells analysed per experiment in the figure legend to improve transparency and reproducibility.

• Line 685 (Methods): It would be helpful to provide the conversion between 1A260 unit in mg/ml and the approximate number of viral particles, to guide readers, particularly since keeping the correct molar ratio is important for controlling labelling efficiency of core preparations.

PLOS authors have the option to publish the peer review history of their article (what does this mean? ). If published, this will include your full peer review and any attached files.

**Do you want your identity to be public for this peer review?** For information about this choice, including consent withdrawal, please see our Privacy Policy .

Reviewer #1: No

Reviewer #2: No

Reviewer #3: No

Reviewer #4: **Yes: ** Alex Borodavka

**Figure resubmission:**
---

## [Editor Report · Decision Letter 1]

19 Aug 2025

Dear Dr. Shmulevitz,

We are pleased to inform you that your manuscript 'Spatiotemporal coordination of reovirus peripheral core replication to perinuclear whole virus assembly' has been provisionally accepted for publication in PLOS Pathogens.

Best regards,

John S.L. Parker, DVM, Ph.D.

Guest Editor

PLOS Pathogens

Sonja Best

Section Editor

PLOS Pathogens

Sumita Bhaduri-McIntosh

Editor-in-Chief

PLOS Pathogens

orcid.org/0000-0003-2946-9497

Michael Malim

Editor-in-Chief

PLOS Pathogens

orcid.org/0000-0002-7699-2064

The authors have addressed all of the reviewers comments.
---

## [Editor Report · Acceptance letter]

Dear Dr. Shmulevitz,

We are delighted to inform you that your manuscript, "Spatiotemporal coordination of reovirus peripheral core replication to perinuclear whole virus assembly," has been formally accepted for publication in PLOS Pathogens.

Best regards,

Sumita Bhaduri-McIntosh

Editor-in-Chief

PLOS Pathogens

orcid.org/0000-0003-2946-9497

Michael Malim

Editor-in-Chief

PLOS Pathogens

orcid.org/0000-0002-7699-2064